# DreamSparse: Escaping from Plato's Cave with 2D Diffusion Model Given Sparse Views

**Paul Yoo**      **Jiaxian Guo**[*]      **Yutaka Matsuo**      **Shixiang Shane Gu**

The University of Tokyo

{paulyoo, jiaxian.guo}@weblab.t.u-tokyo.ac.jp

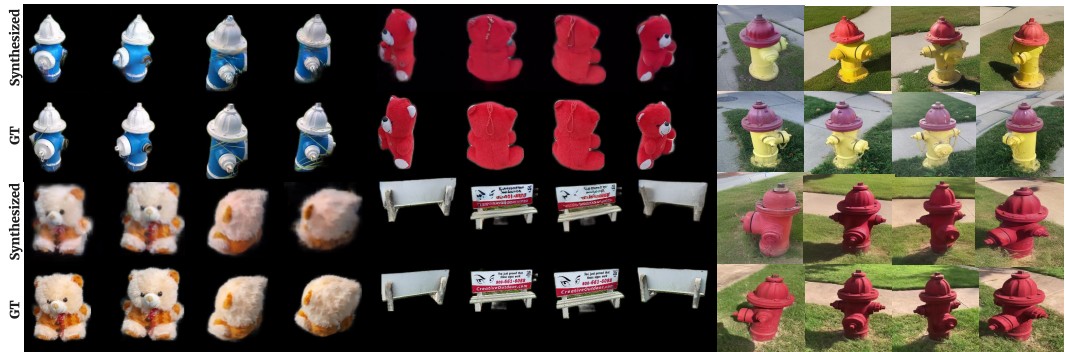

Figure 1: Qualitative results on novel view synthesis of real-world objects from the CO3D dataset.

## Abstract

Synthesizing novel view images from a few views is a challenging but practical problem. Existing methods often struggle with producing high-quality results or necessitate per-object optimization in such few-view settings due to the insufficient information provided. In this work, we explore leveraging the strong 2D priors in pre-trained diffusion models for synthesizing novel view images. 2D diffusion models, nevertheless, lack 3D awareness, leading to distorted image synthesis and compromising the identity. To address these problems, we propose *DreamSparse*, a framework that enables the frozen pre-trained diffusion model to generate geometry and identity-consistent novel view images. Specifically, DreamSparse incorporates a geometry module designed to capture features about spatial information from sparse views as a 3D prior. Subsequently, a spatial guidance model is introduced to convert rendered feature maps as spatial information for the generative process. This information is then used to guide the pre-trained diffusion model to encourage the synthesis of geometrically consistent images without further tuning. Leveraging the strong image priors in the pre-trained diffusion models, DreamSparse is capable of synthesizing high-quality novel views for both object and object-centric scene-level images and generalising to open-set images. Experimental results demonstrate that our framework can effectively synthesize novel view images from sparse views and outperforms baselines in both trained and open-set category images. More results can be found on our project page: `https://sites.google.com/view/dreamsparse-webpage`.

---

[*]Correspondence to Jiaxian Guo <jiaxian.guo@weblab.t.u-tokyo.ac.jp>

37th Conference on Neural Information Processing Systems (NeurIPS 2023).

# 1 Introduction

> "How could they see anything but the shadows if they were never allowed to move their heads?"
> - Plato's Allegory of the Cave

Plato's Allegory of the Cave raises a thought-provoking question about our perception of reality. Human perception of 3D objects is often limited to the projection of the world as 2D observations. We rely on our prior experiences and imagination abilities to infer the unseen views of objects from these 2D observations. As such, perception is to some degree a creative process retrieving from imagination. Recently, Neural Radiance Fields (NeRF) [29] exhibited impressive results on novel view synthesis by utilizing implicit functions to represent volumetric density and color data. However, NeRF requires a large amount of images from different camera poses and additional optimizations to model the underlying 3D structure and synthesize an object from a novel view, limiting its use in real-world applications such as AR/VR and autonomous driving. In most practical applications, typically only a few views are available for each object, in which case leads NeRF to output degenerate solutions with distorted geometry [33, 70, 72].

Recent works [76, 33, 70, 72, 17, 7, 66, 20] started to explore sparse-view novel view synthesis, specifically focusing on generating novel views from a limited number of input images (typically 2-3) with known camera poses. Some of them [33, 70, 72, 17, 7] introduce additional priors into NeRF, *e.g.* depth information, to enhance the understanding of 3D structures in sparse-view scenarios. However, due to the limited information available in few-view settings, these methods struggle to generate clear novel images for unobserved regions. To address this issue, SparseFusion [76] and GeNVS[4] propose learning a diffusion model as an image synthesizer for inferring high-quality novel-view images and leveraging prior information from other images within the same category. Nevertheless, since the diffusion model is only trained within a single category, it faces difficulties in generating objects in unseen categories and needs further distillation for each object, rendering it still impractical.

In this paper, we investigate the utilization of 2D image priors from pre-trained diffusion models, such as Stable Diffusion [41], for generalizable novel view synthesis **without** further per-object training based on sparse views. However, since pre-trained diffusion models are not designed for 3D structures, directly applying them can result in geometrically and textually inconsistent images, compromising the object's identity in Figure 6. To address this issue, we introduce *DreamSparse*, a framework designed to leverage the 2D image prior from pre-trained diffusion models for novel view image synthesis using a few (2) views. In order to inject 3D information into the pre-trained diffusion model and enable it to synthesize images with consistent geometry and texture, we initially employ a geometry module [57] as a 3D geometry prior inspired by previous geometry-based works [40, 39, 31, 23, 70, 57, 20], which is capable of aggregating feature maps across multi-view context images and learning to infer the spatial features, *i.e.* features with spatial information, for the novel view image synthesis. This 3D prior allows us to render an estimate from a previously unseen viewpoint while maintaining accurate geometry.

However, due to the modality gap, the extracted spatial features cannot be directly used as the input to the pre-trained diffusion model for synthesizing geometrically consistent novel view images. Alternatively, we propose a spatial guidance module which is able to convert rendered feature maps into meaningful guidance to change the spatial features [74, 63, 3] in the pre-trained diffusion model, thus promoting the pre-trained diffusion model to generate geometry-consistent novel view image without altering its parameters. Nevertheless, the spatial guidance from feature maps alone cannot completely overcome the hallucination problem of the pre-trained models, as the spatial information encoded in spatial features is limited. This means it cannot guarantee identity consistency in synthesized novel view images. To overcome the limitation, we further propose a noise perturbation method, where we denoise a blurry target view color estimate modulated with noise instead of a randomly initialized noise with the pre-trained diffusion model, so that we can further utilize the coarse shape and appearance information present in 3D geometry model's estimate. In this way, the frozen pre-trained diffusion model is able to effectively synthesize high-quality novel view images with improved consistency in both geometry and identity.

With the strong image synthesis capacity of the frozen pre-trained diffusion model, our approach offers several benefits: **1.** The ability to infer unseen regions of objects without additional training, as pre-trained diffusion models already possess strong image priors learned from large-scale image-text

datasets. **2.** A strong generalization capability, allowing the generation of images across various categories and even in-the-wild images using the strong image priors in the pre-trained diffusion models. **3.** The ability to synthesize high-quality and even object-centric scene-level images without additional per-object optimization. **4.** Since we do not modify the parameters or replace the textual embedding [24] of the pre-trained text-to-image diffusion model, the textual control capability of the pre-existing model is preserved. This allows us to alter the style/texture of the synthesized novel view image with textual control. The comparisons with other methods are given in Table 1.

In our experiments, we applied our framework to the real-world CO3D dataset [37]. The extensive qualitative and quantitative results demonstrated that our approach outperformed baselines in both object-level and object-centric scene-level novel view synthesis settings by a large margin (about 50% in FID and 20% in LPIPS). Specifically, the results in open-set categories of DreamSparse can even achieve competitive performance with those of the baselines in training domains, demonstrating the advantage of exploiting prior from pre-trained 2D diffusion model in open-set generalization.

Table 1: Comparisons with prior works on 1) works with sparse (2-6) input views, 2) hallucinates unseen regions, 3) generalizes to instances in unseen categories because of the pre-trained backbones, and 4) free of training during inference time for novel view synthesis. 5) The ability to edit with textual control.

| | RegNeRF[33] | VolSDF[73] | NeRS[72] | IBRNet[66] | NF[37] | LFN[52] | SRT[47] | PixelNerf[70] | GPNR[57] | VF[20] | 3DFuse[50] | SF[76] | 3DiM[67] | Zero-1-to-3[24] | GeNVS[4] | **Ours** |
|---|---|---|---|---|---|---|---|---|---|---|---|---|---|---|---|---|
| 1) Sparse-Views | ✓ | ✓ | ✓ | ✗ | ✓ | ✓ | ✓ | ✓ | ✗ | ✓ | ✓ | ✓ | ✓ | ✓ | ✓ | ✓ |
| 2) Generate Unseen | ✗ | ✗ | ✗ | ✗ | ✗ | ✗ | ✗ | ✗ | ✗ | ✓ | ✓ | ✓ | ✓ | ✓ | ✓ | ✓ |
| 3) Open-Set Generalization | ✗ | ✗ | ✗ | ✓ | ✓ | ✗ | ✓ | ✓ | ✓ | ✓ | ✓ | ✗ | ✗ | ✓ | ✗ | ✓ |
| 4) Train-Free for NVS | ✗ | ✗ | ✗ | ✓ | ✓ | ✗ | ✓ | ✓ | ✓ | ✓ | ✗ | ✗ | ✓ | ✓ | ✓ | ✓ |
| 5) Textual Control | ✗ | ✗ | ✗ | ✗ | ✗ | ✗ | ✗ | ✗ | ✗ | ✗ | ✓ | ✗ | ✗ | ✗ | ✗ | ✓ |

## 2 Related Works

**Geometry-based Novel View Synthesis.** Prior research on Novel View Synthesis (NVS) largely focuses on recovering the 3D structure of a scene. This is achieved by estimating the parameters of the input images' cameras and subsequently applying a multi-view stereo (MVS) technique, as indicated by several studies [55, 48, 11, 1]. These methods use explicit geometry proxies to facilitate NVS. However, it often fails to synthesize novel views that are both photo-realistic and comprehensive, particularly in the case of occluded areas. In order to address this issue, recent strategies [39, 40] have attempted to integrate the 3D geometry derived from an MVS pipeline with NVS approaches based on deep learning. Despite its progress, the overall quality may deteriorate if the MVS pipeline encounters failures. The utilization of other explicit geometric representations has also been explored by various recent NVS techniques. These include the usage of depth maps [10, 62], multi-plane images [9, 78], or voxels [53, 25]. While geometry-based methods have advanced novel view synthesis [29], their limited capacity to model uncertainties in unseen regions and the lack of a 2D prior often restrict their image synthesis quality and generalization for unseen category objects. In contrast, our approach seeks to meld the strengths of both geometry-based methods and the robust 2D pre-trained diffusion model. This fusion showcases our method's prowess in synthesizing images that are not only geometry-consistent but also of high quality, ultimately leading to enhanced generalization performance on objects in unseen categories.

**Sparse-view Novel View Synthesize.** Novel View Synthesis (NVS) from few views aims to generate a new image from a novel viewpoint using a limited number of 2D images [59]. Because of the limited information available in this setting, many prior works [60, 79, 54, 18, 51, 58] necessitate per-object or per-category test-time optimization, which makes them impractical. Other efforts [62, 34, 58, 70, 61, 6, 66, 20, 54, 43, 38, 17, 57, 14, 13] learn a latent prior over shapes or encode observations into feature representations and subsequently render novel views via volume rendering. In order to hallucinate unseen regions arising from limited input view images, several recent approaches, such as 3DiM [67], SparseFusion [76], NeRDi [7], Zero-1-to-3 [24] and GeNVS

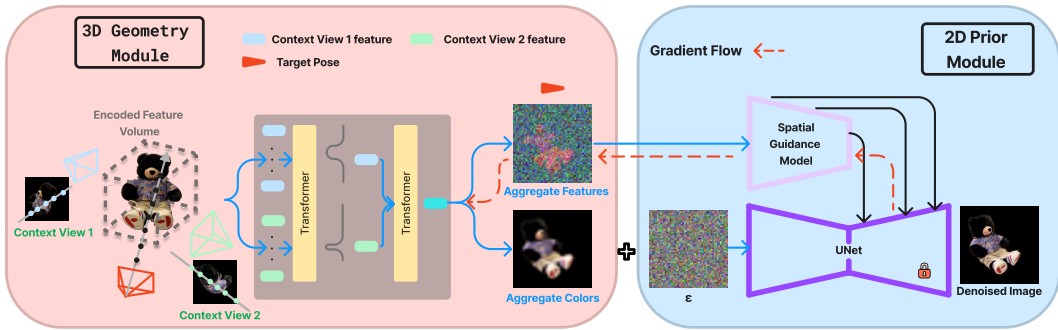

Figure 2: The illustration of the method. The first stage involves utilizing a 3D geometry module to estimate 3D structure and aggregate features from context views. In the next stage, a pre-trained 2D diffusion model conditioned on the aggregate features is leveraged to learn a spatial guidance model that guides the diffusion process for accurate synthesis of the underlying object.

[4], have utilized image diffusion models. On the other hand, we propose a framework to guide a foundational 2D diffusion model while keeping it frozen and simultaneously enforce 3D structure through a geometry-based module.

**Diffusion Model for Novel View Synthesize**    In order to achieve high-quality novel view synthesis, recent works [16, 56, 41, 45, 26, 2, 71, 77, 7, 21, 22, 28, 68, 36, 32, 30, 8] introduce diffusion models conditioned on text or images to generate or reconstruct 3D representations. In the context of novel view synthesis, 3DiM[67] performs view synthesis only conditioned on input images and poses without 3D geometry, leaving vulnerabilities in generating 3D-consistent images. SparseFusion [76] and GeNVS [4] proposed to integrate additional geometric structure as training conditions for the diffusion model, thereby enabling the generation of 3D-consistent images. However, due to the absence of strong 2D prior in the diffusion models they employ, these approaches are challenging to generalize to objects in open-set categories. In contrast, our approach utilizes a frozen diffusion model pre-trained on a large-scale dataset [49], enhancing its generalization ability for objects in open-set categories.

## 3   Method

Given a few context images $\{C_i^{inputs}\}_{i=1}^N$ and their poses $\pi_i$, we aim to leverage the 2D image prior from pre-trained diffusion models to synthesize a novel view image at a target pose $\pi_{target}$. Because pre-trained diffusion models are not 3D-aware, we first employ a geometry-aware module as a 3D prior to extract features in a 3D volume encoded from given context images. In order to leverage feature map renderings for the pre-trained diffusion model, we further propose a spatial guidance model to convert geometry-grounded features into spatial features [63, 3] with consistent shape in diffusion models, guiding it to synthesize novel view images with correct geometry. However, we discover that relying solely on the diffusion model to sample an accurate reconstruction from random noise is inadequate at maintaining the object's identity due to the hallucination problem [19, 36] as shown in 6. Therefore, we propose a noise perturbation method to alleviate it, guiding the diffusion model to synthesize a novel view image with correct geometry and identity. The overall pipeline is illustrated in Fig. 2.

### 3.1   3D Geometry Module

In order to infuse 3D awareness into 2D diffusion models, we propose a geometry module to extract features with geometric information for 2D diffusion models. In order to obtain geometry grounding, the process begins by casting a query ray from $\pi^{target}$ and sampling uniformly spaced points along the ray. For each 3D point, we aim to learn density weighting for computing a weighted linear combination of features along the query ray. Subsequently, this per-ray feature is aggregated across multiple context images, yielding a unified perspective on the 3D structure we aim to reconstruct. Lastly, we render a feature map at $\pi^{target}$ by raycasting from the target view. Next, we will present the details of this model.

**Point-wise density weighting for each context image.** For each input context image $C_i^{inputs}$, our geometry module first extracts semantic features using a ResNet50 [12] backbone and then reshapes the encoded feature into a 4 dimensional volumetric representation $V_i \in \mathbb{R}^{c \times d \times h \times w}$, where $h$ and $w$ are the height and width of the feature volume, respectively, $d$ is the depth resolution, and $c$ is the feature dimension. We pixel-align the spatial dimensions of the volume to that of the original input image via bilinear upsampling. To derive benefit from multi-scale feature representation, we draw feature maps from the first three blocks of the backbone and reshape them into volumetric representations capturing the same underlying 3D space. Given a 3D query point $\boldsymbol{p}_j$ along a query ray $\mathbf{r}^i$, we sample feature vectors from all three scales of feature volumes using trilinear interpolation concatenating them together. To calculate the point-wise density weighting, we employ a transformer [64] with a linear projection layer at last followed by a softmax operation to determine a weighted linear combination of point features, resulting in a per-ray feature vector. Limited by the page limit, we leave further implementation details in the Appendix A.

**Aggregate features from different context images.** To understand the unified structure of the 3D object, we consolidate information from all given context images. More specifically, we employ an extra transformer, enabling us to dynamically consolidate ray features from a varying number of context images that correlate with each query ray. The final feature map rendering at a query view is constructed by raycasting from the query view and computing per-ray feature vector for each ray. We render the feature map $\boldsymbol{F}$ at a resolution of $\mathbb{R}^{64 \times 64}$, compositing features sampled from a 3D volume with geometry awareness with respect to the target view. We denote $g$ as the feature map rendering function and $\boldsymbol{F}$ as the resulting aggregate feature map.

$$\boldsymbol{F} = g_\phi(\boldsymbol{\pi}^{inputs}, \mathbf{C}^{inputs}, \boldsymbol{\pi}^{target}) \tag{1}$$

where $\boldsymbol{F} \in \mathbb{R}^{d \times 64 \times 64}$ with $d = 256$, and $\phi$ is trainable parameters.

**Color Estimation** To enforce geometric consistency, we directly obtain aggregation weights from the transformer outputs and linearly combine RGB color values drawn from the context images to render a coarse color estimate $E$ at the query view.

$$E = g_{\phi,color}(\boldsymbol{\pi}^{inputs}, \mathbf{C}^{inputs}, \boldsymbol{\pi}^{target}) \tag{2}$$

We impose a color reconstruction loss on the coarse image against the ground-truth image.

$$\mathcal{L}_{recon} = \sum_{\boldsymbol{\pi}^{target}} \left\| g_{\phi,color}(\boldsymbol{\pi}^{inputs}, \mathbf{C}^{inputs}, \boldsymbol{\pi}^{target}) - C^{target} \right\|^2 \tag{3}$$

### 3.2 Spatial Guidance Module

Because of the modality gap between the feature map rendering $\boldsymbol{F}$ and the input of the pre-trained diffusion model, feature maps $\boldsymbol{F}$ cannot be directly used as the input of the pre-trained diffusion model. To leverage the 3D information in the feature maps, we propose the spatial guidance module to convert the feature maps into guidance to rectify the spatial features [63, 74, 3] that have a role in forming fine-grained spatial information in the diffusion process (normally the feature maps after the 4-th layer). To derive this guidance from spatial information in the features $\boldsymbol{F}$, we construct our spatial guidance module following ControlNet [74] which trains a separate copy of all the encoder blocks as well as the middle block from Stable Diffusion's U-Net with 1x1 convolution layers initialized with zeros between each block. Let $T_\theta$ be the spatial guidance module, and intermediate outputs from each block $j$ of $T_\theta$ as $T_{\theta,j}(\boldsymbol{F})$ with weight $\lambda$. In order to change the spatial features in the pre-trained diffusion model, we directly add $T_{\theta,j}(\boldsymbol{F})$ into the corresponding decoder block of the pre-trained diffusion model's U-Net. By optimizing $T_\theta$ with gradients backpropagated from the pre-trained diffusion model's noise prediction objective.

$$\mathcal{L}_{diffusion} = \mathbb{E}_{x_0,t,\boldsymbol{F},\epsilon \sim \mathcal{N}(0,1)} \left[ \left\| \epsilon - \epsilon_\psi(x_{t+1}, t, T_\theta(\boldsymbol{F})) \right\|^2 \right] \tag{4}$$

$T_\theta$ will be optimized to learn how to convert the feature map from the geometry module into semantically meaningful guidance to rectify spatial features in the diffusion process, enabling it to generate geometry-consistent images. In Section 4.5, we visualize the spatial features after adding the spatial guidance to show the effects of the spatial guidance model. During training, we jointly optimize $g_\phi$ and $T_\theta$ using the overall loss.

$$\min_{\phi,\theta} \mathcal{L}_{recon}(g_\phi) + \mathcal{L}_{diffusion}(T_\theta) \tag{5}$$

While in training time, we use a ground-truth image as $x_0$ to optimize $\mathcal{L}_{diffusion}$, in inference time, we initialize $x_0$ with an image rendered from $g_{\phi,color}$.

**Noise Perturbation**   While spatial guidance module by itself is able to guide the pre-trained diffusion model to synthesize novel view images with consistent geometry. It still cannot always synthesize images with the same identity as context views because of the hallucinate problem [19, 36] in the pre-trained models. To alleviate this problem, we propose adding noise perturbation to the novel view estimate $E$ from the geometry module and denoising the result with the pre-trained diffusion model, *e.g.* Stable Diffusion [42], so that it can leverage the identity information from the estimate. As shown by [27], applying the denoising procedure can project the sample to a manifold of natural images. We use the formulations from denoising diffusion models [16] to perturb an initial image $x_0 = E$ with Gaussian noise to get a noisy image $x_t$ as follows:

$$x_t = \sqrt{\bar{\alpha}_t}x_0 + \sqrt{1 - \bar{\alpha}_t}\epsilon \tag{6}$$

where $\bar{\alpha}_t$ depends on scheduling hyperparameters and $\epsilon \sim \mathcal{N}(0, 1)$. During the training time, the noise is still randomly initialized, and we use the Noise Perturbation method in the inference time to improve the identity consistency. We show its ablation study in Section 4.5.

## 4   Experiments

In this section, we first validate the efficacy of our DreamSparse framework on zero-shot novel view synthesis by comparing it with other baselines. Then, we perform ablation studies on important design choices, such as noise perturbation and visualization of spatial features, to understand their effects. We also present qualitative examples of our textual control ability and include a discussion on observations.

### 4.1   Dataset and Training Details

Following SparseFusion [76], we perform experiments on real-world scenes from the Common Objects in 3D (CO3Dv2) [37], a dataset with real-world objects annotated with camera poses. We train and evaluate our framework on the CO3Dv2 [37] dataset's `fewview_train` and `fewview_dev` sequence sets respectively. We use Stable Diffusion v1.5 [42] as the frozen pre-trained diffusion model and DDIM [56] to synthesize novel views with 20 denoising steps. The resolutions of the feature map for the spatial guidance module and latent noise are set as $64 \times 64$ with spatial guidance weight $\lambda = 2$. The three transformers used in the geometry module all contain 4 layers, and the output feature map is rendered at a resolution of $64 \times 64$ to match the latent noise dimensions. We jointly train the geometry and the spatial modules on 8 A100-40GB GPUs for 3 days with a batch size of 15. To demonstrate our framework's generalization capability at object-level novel view synthesis, we trained our framework on a subset of 10 categories as specified in [37]. During each training iteration, a query view and one to four context views of an object were randomly sampled as inputs to the pipeline. To further evaluate scene-level novel view synthesis capability, we trained our framework on the hydrant category, incorporating the full background, using the same training methodology as above.

### 4.2   Competing Methods

We compare against previous state-of-the-art (SoTA) methods for which open-source code is available. We have included PixelNeRF [69], a feature re-projection method, in our comparison. Additionally, we compare our methods against SparseFusion [76], the most recently published SoTA method that utilizes a diffusion model for NVS. We train our framework and SparseFusion on 10 categories of training sets. The PixelNeRF training was conducted per category due to its category-specific hyperparameters. For a fair comparison, all methods perform NVS **without** per-object optimization during the inference time. Because we do not replace the textual embedding in the pre-trained diffusion model, we use the prompt 'a picture of <class_name>' as the default prompt for both training and inference.

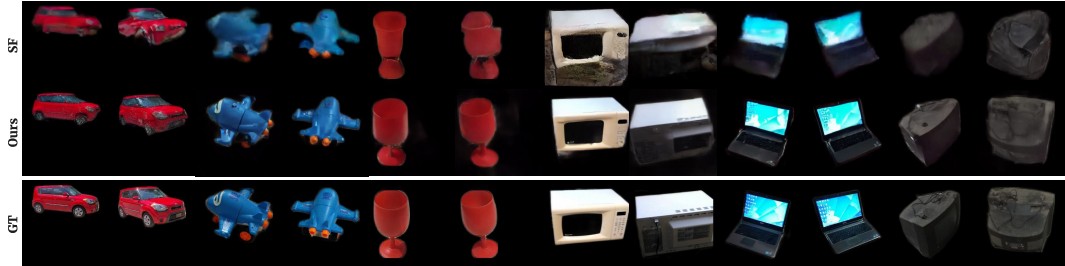

Figure 3: Novel view synthesizing results on **open-set** category objects with the same context image inputs, where SF denotes SparseFusion [76] and GT denotes Ground-Truth image. More results are given at our project webpage and appendix F.

Table 2: Quantitative evaluation metrics on 10 subset of training categories from CO3D, where PN denotes PixelNerf [69] and SF denotes SparseFusion [76]. Limited by the width, we only show FID and LPIPS score here.

|  | Apple | | Ball | | Bench | | Cake | | Donut | | Hydrant | | Plant | | Suitcase | | Teddybear | | Vase | | Avg. | |
|---|---|---|---|---|---|---|---|---|---|---|---|---|---|---|---|---|---|---|---|---|---|---|
|  | FID | LPIPS | FID | LPIPS | FID | LPIPS | FID | LPIPS | FID | LPIPS | FID | LPIPS | FID | LPIPS | FID | LPIPS | FID | LPIPS | FID | LPIPS | FID | LPIPS |
| PN | 247.1 | 0.57 | 319.2 | 0.56 | 344.0 | 0.53 | 380.8 | 0.58 | 340.8 | 0.63 | 318.7 | 0.48 | 335.0 | 0.52 | 333.9 | 0.45 | 352.1 | 0.56 | 288.9 | 0.47 | 326.1 | 0.54 |
| SF | 110.9 | 0.28 | 143.5 | 0.30 | 255.8 | 0.34 | 185.7 | 0.33 | 126.6 | 0.29 | 165.7 | 0.23 | 168.5 | 0.31 | 202.6 | 0.28 | 199.3 | 0.34 | 167.8 | 0.23 | 172.6 | 0.29 |
| Ours | **42.8** | **0.19** | **45.8** | **0.19** | **122.5** | **0.25** | **105.2** | **0.23** | **67.2** | **0.21** | **87.8** | **0.16** | **86.2** | **0.23** | **100.2** | **0.20** | **91.2** | **0.23** | **69.1** | **0.16** | **81.8** | **0.21** |

## 4.3 Main Results Analysis

Given 2 context views, we evaluate novel view synthesis quality using the following metrics: FID [15], LPIPS [75], and PSNR [2]. We believe that the combination of FID, LPIPS, and PSNR provides a comprehensive evaluation of novel view synthesis quality. FID and LPIPS measure the perceptual quality of the images, while PSNR measures the per-pixel accuracy. We note that PSNR has some drawbacks as a metric for evaluating generative models. Specifically, PSNR tends to favor blurry images that lack detail. This is because PSNR only measures the per-pixel accuracy of an image, and does not take into account the overall perceptual quality of the image. By using all three metrics, we can get a more complete picture of the quality of the images generated by our model.

### 4.3.1 Object Level Novel View Synthesis

**In-Domain Evaluation** We evaluate the performance of unseen objects NVS in training 10 categories. The quantitative results are presented in Table 5, which clearly demonstrates that our method surpasses the baseline methods in terms of both FID and LPIPS metrics. More specifically, DreamSparse outperforms SparseFusion by a substantial margin of 53% in the FID score and 28% in LPIPS. This significant improvement can be attributed to DreamSparse's capacity to generate sharper, semantically richer images, as depicted in Figure 1. This indicates the benefits of utilizing the potent image synthesis capabilities of pre-trained diffusion models.

**Open-Set Evaluation** We also evaluate the performance of objects NVS in open-set 10 categories, because PixelNerf is per-category trained, we do not report its open-set generalization results. According to Table 6, it is evident that our method surpasses the baseline in both evaluation metrics in all categories, surpassing the second-best method by 28% in LPIPS and 43% in FID. Moreover, the results derived from our method are not just competitive, but can even be compared favourably to the training category evaluations of the baseline in Table 5 (122.2 vs 172.2 in FID and 0.24 vs 0.29 in LPIPS). This clearly illustrates the benefits of utilizing 2D priors from a large-scale, pre-trained 2D diffusion model for open-set generalization. We also show the qualitative results in Figure 3, and it shows that the novel view image synthesised by our method can still achieve sharp and meaningful results on objects in open-set categories.

Table 3: Quantitative evaluation on 10 **open-set** categories outside of the training split, where SF denotes SparseFusion [76].

| | Bicycle | | Car | | Couch | | Laptop | | Microwave | | Motorcycle | | Bowl | | Toyplane | | TV | | Wineglass | | Avg. | |
|---|---|---|---|---|---|---|---|---|---|---|---|---|---|---|---|---|---|---|---|---|---|---|
| | FID | LPIPS | FID | LPIPS | FID | LPIPS | FID | LPIPS | FID | LPIPS | FID | LPIPS | FID | LPIPS | FID | LPIPS | FID | LPIPS | FID | LPIPS | FID | LPIPS |
| SF | 217.7 | 0.34 | 209.5 | 0.30 | 201.1 | 0.44 | 223.8 | 0.36 | 200.4 | 0.40 | 205.8 | 0.35 | 198.4 | 0.26 | 202.2 | 0.26 | 261.4 | 0.33 | 208.5 | 0.23 | 212.9 | 0.33 |
| Ours | **154.7** | **0.26** | **116.5** | **0.22** | **129.9** | **0.31** | **167.3** | **0.26** | **127.2** | **0.29** | **115.4** | **0.29** | **61.9** | **0.16** | **98.5** | **0.19** | **167.9** | **0.24** | **82.7** | **0.15** | **122.2** | **0.24** |

Table 4: Quantitative evaluation metrics for object-centric scene-level novel view synthesis on the hydrant category from CO3D, where SF denotes SparseFusion [76].

| | 1 View | | | 2 Views | | | 5 Views | | |
|---|---|---|---|---|---|---|---|---|---|
| | FID↓ | LPIPS↓ | PSNR↑ | FID↓ | LPIPS↓ | PSNR↑ | FID↓ | LPIPS↓ | PSNR↑ |
| PixelNeRF [69] | 343.89 | 0.75 | **13.31** | 319.96 | 0.74 | **13.94** | 286.30 | 0.71 | 14.59 |
| SparseFusion [76] | 272.72 | 0.81 | 13.05 | 255.05 | 0.78 | 13.55 | 231.73 | 0.71 | **14.91** |
| Ours | **75.63** | **0.59** | 13.02 | **73.47** | **0.56** | 13.48 | **70.62** | **0.54** | 14.15 |

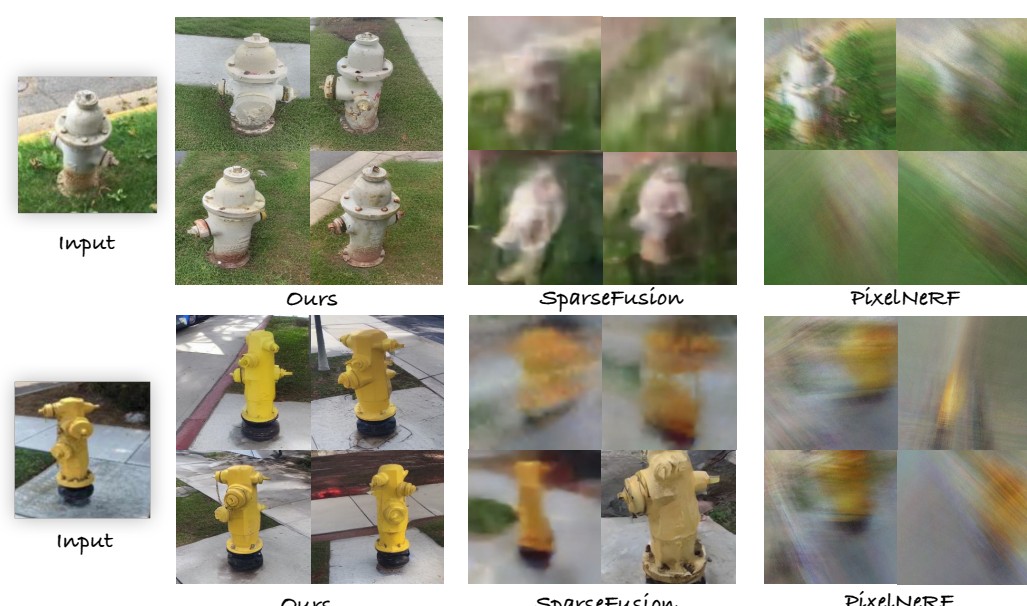

Figure 4: Qualitative results of Object-Centric Scene Level novel view synthesis outputs from all baselines.

### 4.3.2 Object-Centric Scene Novel View Synthesis

We report our evaluation results on object-centric scene-level NVS in Table 7. As shown in the table, DreamSparse significantly outperforms the baselines in terms of FID and LPIPS scores, surpassing the second-best performance by approximately 70% in FID and 24% in LPIPS, respectively. This underscores the effectiveness of our method in the context of object-centric scene NVS tasks. Despite our method showing comparable performance to the baseline in terms of Peak Signal-to-Noise Ratio (PSNR), it's worth mentioning that PSNR often favors blurry images lacking in detail [46, 44, 4]. This becomes evident in Figure 4, where despite our sharp and consistent synthesis results, PSNR still leans towards the blurry image produced by PixelNeRF.

### 4.4 Textual Control Style Transfer

As we do not replace or remove text conditioning in the pre-trained diffusion model, our method is additionally capable of controlling the image generation with text. We demonstrate an example use case where we conduct both novel view synthesis and style transfer via text in Figure 5.

---

[2] https://en.wikipedia.org/wiki/Peak_signal-to-noise_ratio

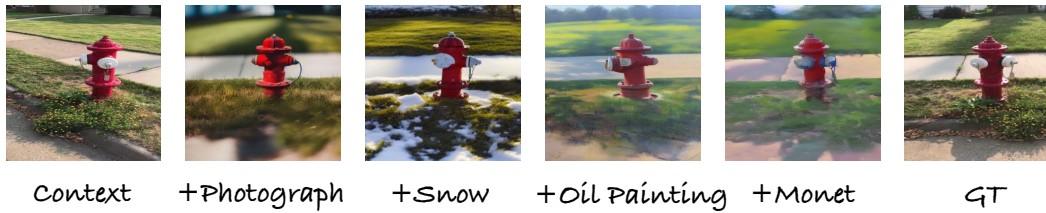

Context      +Photograph      +Snow      +Oil Painting      +Monet      GT

Figure 5: Qualitative results of novel view synthesis with textual control style transfer

## 4.5 Ablation Studies

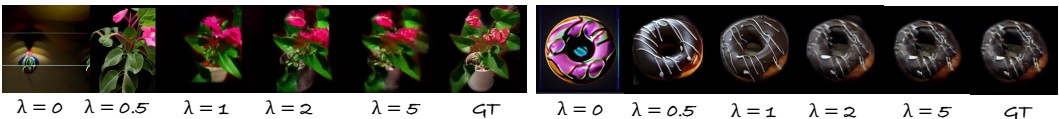

$\lambda=0$   $\lambda=0.5$   $\lambda=1$   $\lambda=2$   $\lambda=5$   GT      $\lambda=0$   $\lambda=0.5$   $\lambda=1$   $\lambda=2$   $\lambda=5$   GT

Figure 6: Effect of the spatial guidance weighting of spatial guidance signals at the residual connections in the middle and decoder blocks of the diffusion model.

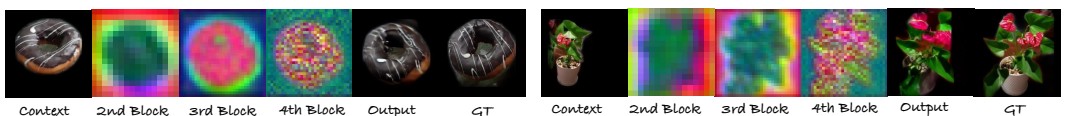

Context   2nd Block   3rd Block   4th Block   Output   GT      Context   2nd Block   3rd Block   4th Block   Output   GT

Figure 7: The spatial feature visualization with spatial guidance model, where context denotes the input context image, 2nd block denotes the visualization of feature maps from the 2nd block of the decoder and output denotes the synthesised novel view image.

**Number of Input Views** We investigate the performance by varying the number of context view inputs. The results are depicted in Table 6, which clearly illustrates the enhancement in three evaluation metrics as the number of input views increases. Moreover, the performance disparity between the **single view** and multiple input views is less pronounced in our method than in other baselines - a 6% difference vs 14% difference in Sparse Fusion vs a 17% difference in PixelNerf. This observation leads to two key conclusions: 1) DreamSparse exhibits greater robustness in response to variations in the number of context view inputs. 2) Despite the decrease in performance, DreamSparse can efficiently synthesise novel views from a single input view.

**Spatial Feature Visualization** To investigate the impact of the spatial guidance model, we employ Principal Component Analysis (PCA) [35] to visualize the spatial features post-integration of spatial guidance following [63]. As shown in Figure 7, the visualized feature maps from the 2nd, 3rd, and 4th blocks of the UNet decoder indicate that despite the contextual view's geometry varying from that of the novel view, the feature maps steered by our spatial guidance model maintain alignment with the geometry of the ground truth image. This consistency enables the pre-trained diffusion model to generate images that accurately mirror the original geometry.

**Spatial Guidance Weight** We investigate the effects of spatial guidance weight on the quality and consistency of synthesized novel view images. Our study varies the spatial guidance weight $\lambda$, and the results in Fig 6 showed that when $\lambda=0$ (indicating no spatial guidance), the pre-trained diffusion model failed to synthesize a novel view image that was consistent in terms of geometry and identity. However, as the weight increased, the synthesized images exhibited greater consistency with the ground truth. It is important to note, though, that an excessively high weight could diminish the influence of features in the pre-trained diffusion model, potentially leading to blurry output. Given the trade-off between quality and consistency, we set $\lambda=2$ as the default hyperparameter.

**Effect of Noise Perturbing Color Estimation** The impact of the Noise Perturbation method is showcased in Figure 8. It is evident that when the diffusion process begins from random noise, the spatial guidance model can successfully guide the pre-trained diffusion model to synthesize images

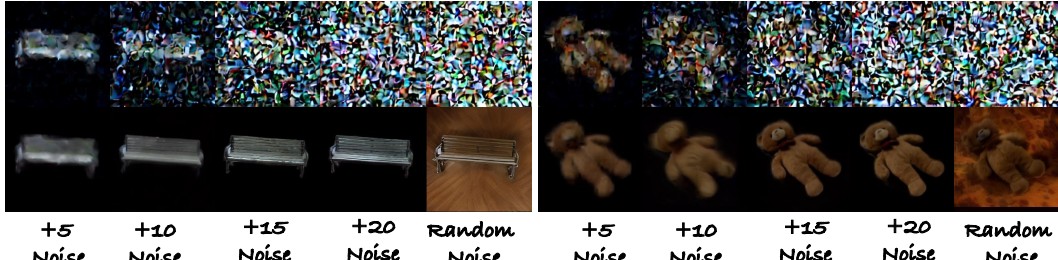

Figure 8: Effect of noise perturbation level on novel view image synthesis. The images in the top row are visualizations of noised images as input to the diffusion process, and the images in the bottom row are the synthesized images from diffusion model. +5 noise denotes the addition of 5 steps of noise to the color estimate from the geometry model, and random noise denotes randomly sampled Gaussian noise. All images were generated using a DDIM [56] sampler with 20 inference steps.

with consistent geometry. However, the color or illumination density information is partially lost, leading to distortions in the synthesized novel view. In contrast, synthesizing the image from the noise that is added to the color estimation in the geometry model yields better results. As depicted in '+20 Noise' in Figure 8, the pre-trained diffusion model can effectively utilize the color information in the estimates, resulting in a more consistent image synthesis. We also experimented with varying the noise level added to the estimate. Our observations suggest that if the noise added to the blurry estimation is insufficient, the pre-trained diffusion model struggles to denoise the image because of the distribution mismatch between the blurry color estimate and Gaussian distribution, thereby failing to produce a sharp and consistent output.

## 5    Conclusion

In this paper, we present the *DreamSparse*, a framework that leverages the strong 2D priors of a frozen pre-trained text-to-image diffusion model for novel view synthesis from sparse views. Our method outperforms baselines on existing benchmarks in both training and open-set object-level novel view synthesis. Further results corroborate the benefits of utilizing a pre-trained diffusion model in object-centric scene NVS as well as in the generation of text-controlled scenes style transfer, clearly outperforming existing models and demonstrating the potential of leveraging the 2D pre-trained diffusion models for novel view synthesis.

**Limitations and Negative Social Impact**    Despite its capabilities, we discovered that our 3D Geometry Module struggles with generating complex scenes, especially ones with non-standard geometry or intricate details. This is due to the limited capacity of the geometry module and limited data, and we will introduce a stronger geometry backbone and train it on larger datasets. On the social impact front, our technology could potentially lead to job displacement in certain sectors. For instance, professionals in fields such as graphic design or 3D modelling might find their skills becoming less in-demand as AI-based techniques become more prevalent and advanced. It's important to note that these negative implications are not exclusive to this study, and should be widely considered and addressed within the realm of AI research.

## 6    Acknowledgement

Computational resources of AI Bridging Cloud Infrastructure (ABCI) provided by National Institute of Advanced Industrial Science and Technology (AIST) were used for the experiments.

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

# Supplementary Material of "DreamSparse: Escaping from Plato's Cave with 2D Diffusion Model Given Sparse Views"

**Paul Yoo**     **Jiaxian Guo**[‡]     **Yutaka Matsuo**     **Shixiang Shane Gu**
The University of Tokyo
{paulyoo, jiaxian.guo}@weblab.t.u-tokyo.ac.jp

To enhance visual presentation, we have showcased more qualitative samples on our paper's website. `https://sites.google.com/view/dreamsparse-webpage`. Next, we present the details of our method and additional quantitative comparisons with a recent baseline GBT and an ablation study on the 3D geometry prior module.

## A  Method Implementation Details

**Feature Extraction Backbone.**    To encode a volumetric feature field, we input a context image through a ResNet50 [12] backbone and extract multi-scale feature maps from the first three blocks having feature dimensions 256, 512, and, 1024 respectively. To accommodate an additional depth dimension, the feature dimension is divided by the depth, which we set to 64, resulting in features of size 4, 8, 16 for the three volumes respectively. The final feature vector at a query 3D point is a concatenation of features from all three volume scales, making the final feature vector dimension 28.

**Weighting Points Along a Ray.**    We employ a Transformer [64] to learn weightings for computing a linear combination of features of points along a given query ray. To form the input sequence for the Transformer, we follow the input parameterization of [57]. For each point, we concatenate the final feature vector from the backbone with an encoding of the query point and depth along the ray as a positional cue. The query point encoding is computed by first extracting relative poses for all context views with respect to the target view and representing the ray from each context camera origin to the query point in Plücker coordinates. The depth of each point along the ray is additionally parameterized by a sinusoidal positional encoding as in [29]. The output sequence of the Transformer is followed by a linear projection layer and a Softmax operation to yield scalar densities for computing a weighted sum of output features corresponding to points on a query ray.

**Multi-view Aggregation Transformer.**    Once per-ray feature aggregate for each context view has been computed, we similarly learn to combine across all context views via a Transformer. We again concatenate per-ray feature vector output from the previous step with the same Plücker parameterization of query point and sinusoidal positional encoding of depth. All hidden and output dimensions of the Transformers are set to 256.

## B  Table Update with PSNR and an Additional Baseline

For completeness, we make an addition to the quantitative results to reflect PSNR measurements. Furthermore, we include another baseline GBT [65], a geometry-biased Transformer-based method for novel view synthesis from sparse context views, that demonstrates robust results on 10 categories of CO3D. Although GBT acheives comparable PSNR across training and open-set categories, synthesized novel views tend to be blurry especially for open-set categories and thus underperforming in FID and LPIPS.

---

[*]Correspondence to Jiaxian Guo <jiaxian.guo@weblab.t.u-tokyo.ac.jp>

[‡]Correspondence to Jiaxian Guo <jiaxian.guo@weblab.t.u-tokyo.ac.jp>

Table 5: Quantitative evaluation metrics on 10 subset of training categories from CO3D, where PN denotes PixelNerf [69], SF denotes SparseFusion [76], and Geom Prior is the geometry prior module from our method.

| | Apple | | | Ball | | | Bench | | | Cake | | | Donut | | | Hydrant | | |
|---|---|---|---|---|---|---|---|---|---|---|---|---|---|---|---|---|---|---|
| | FID ↓ | LPIPS↓ | PSNR ↑ | FID ↓ | LPIPS↓ | PSNR ↑ | FID ↓ | LPIPS↓ | PSNR ↑ | FID ↓ | LPIPS↓ | PSNR ↑ | FID ↓ | LPIPS↓ | PSNR ↑ | FID↓ | LPIPS↓ | PSNR ↑ |
| PN | 247.1 | 0.57 | 14.76 | 319.2 | 0.56 | 14.25 | 344.0 | 0.53 | 15.16 | 380.8 | 0.58 | 15.10 | 340.8 | 0.63 | 15.76 | 318.7 | 0.48 | 15.20 |
| GBT | 168.85 | 0.27 | 22.96 | 175.18 | 0.28 | 21.45 | 324.28 | 0.33 | 19.10 | 264.86 | 0.32 | 20.71 | 209.33 | 0.29 | 22.78 | 237.22 | 0.23 | 21.82 |
| SF | 110.9 | 0.28 | 20.91 | 143.5 | 0.30 | 20.25 | 255.8 | 0.34 | 17.21 | 185.7 | 0.33 | 19.39 | 126.6 | 0.29 | 20.62 | 165.7 | 0.23 | 19.35 |
| Ours (Geom Prior) | 123.98 | 0.33 | 22.96 | 134.41 | 0.33 | 22.09 | 238.95 | 0.38 | **19.42** | 197.00 | 0.35 | 21.41 | 162.12 | 0.34 | **22.95** | 211.03 | 0.26 | **21.90** |
| Ours | **42.8** | **0.19** | **23.72** | **45.8** | **0.19** | **23.12** | **122.5** | **0.25** | 18.68 | **105.2** | **0.23** | **21.73** | **67.2** | **0.21** | 22.88 | **87.8** | **0.16** | 21.71 |

| | Plant | | | Suitcase | | | Teddybear | | | Vase | | | Avg. | | |
|---|---|---|---|---|---|---|---|---|---|---|---|---|---|---|---|
| | FID ↓ | LPIPS↓ | PSNR ↑ | FID ↓ | LPIPS↓ | PSNR ↑ | FID ↓ | LPIPS↓ | PSNR ↑ | FID ↓ | LPIPS↓ | PSNR ↑ | FID ↓ | LPIPS↓ | PSNR ↑ |
| PN | 335.0 | 0.52 | 18.08 | 333.9 | 0.45 | 20.12 | 352.1 | 0.56 | 14.85 | 288.9 | 0.47 | 15.91 | 326.1 | 0.54 | 15.91 |
| GBT | 254.74 | 0.29 | 21.29 | 283.38 | 0.28 | 23.41 | 294.84 | 0.32 | 19.93 | 255.22 | 0.26 | 22.28 | 246.79 | 0.29 | 21.57 |
| SF | 168.5 | 0.31 | 19.60 | 202.6 | 0.28 | 21.87 | 199.3 | 0.34 | 18.03 | 167.8 | 0.23 | 21.36 | 172.6 | 0.29 | 19.85 |
| Ours (Geom Prior) | 206.21 | 0.35 | **21.64** | 210.56 | 0.30 | **23.69** | 223.82 | 0.36 | **20.68** | 204.06 | 0.27 | 22.74 | 191.21 | 0.33 | 21.95 |
| Ours | **86.2** | **0.23** | 20.59 | **100.2** | **0.20** | 23.30 | **91.2** | **0.23** | 20.51 | **69.1** | **0.16** | **23.03** | **81.8** | **0.21** | **22.03** |

Table 6: Quantitative evaluation on 10 **open-set** categories outside of the training split, where SF denotes SparseFusion [76].

| | Bicycle | | | Car | | | Couch | | | Laptop | | | Microwave | | | Motorcycle | | |
|---|---|---|---|---|---|---|---|---|---|---|---|---|---|---|---|---|---|---|
| | FID ↓ | LPIPS↓ | PSNR ↑ | FID ↓ | LPIPS↓ | PSNR ↑ | FID ↓ | LPIPS↓ | PSNR ↑ | FID ↓ | LPIPS↓ | PSNR ↑ | FID ↓ | LPIPS↓ | PSNR ↑ | FID ↓ | LPIPS↓ | PSNR ↑ |
| SF | 217.7 | 0.34 | 17.57 | 209.5 | 0.30 | 17.49 | 201.1 | 0.44 | 19.25 | 223.8 | 0.36 | 18.73 | 200.4 | 0.40 | 17.65 | 205.8 | 0.35 | 17.52 |
| GBT | 259.77 | 0.33 | 19.10 | 287.64 | 0.33 | 18.18 | 272.82 | 0.43 | 20.10 | 276.15 | 0.36 | 19.75 | 276.44 | 0.41 | 17.40 | 294.84 | 0.35 | 18.50 |
| Ours (Geom Prior) | 235.36 | 0.38 | **19.39** | 218.86 | 0.34 | **19.11** | 182.56 | 0.39 | **21.58** | 231.58 | 0.39 | **21.09** | 213.55 | 0.45 | **19.26** | 259.89 | 0.41 | **18.65** |
| Ours | **154.7** | **0.26** | 18.16 | **116.5** | **0.22** | 18.38 | **129.9** | **0.31** | 20.36 | **167.3** | **0.26** | 20.13 | **127.2** | **0.29** | 18.96 | **115.4** | **0.29** | 17.35 |

| | Bowl | | | Toyplane | | | TV | | | Wineglass | | | Avg. | | |
|---|---|---|---|---|---|---|---|---|---|---|---|---|---|---|---|
| | FID ↓ | LPIPS↓ | PSNR ↑ | FID ↓ | LPIPS↓ | PSNR ↑ | FID ↓ | LPIPS↓ | PSNR ↑ | FID ↓ | LPIPS↓ | PSNR ↑ | FID ↓ | LPIPS↓ | PSNR ↑ |
| SF | 198.4 | 0.26 | 19.08 | 202.2 | 0.26 | 18.81 | 261.4 | 0.33 | 22.58 | 208.5 | 0.23 | 19.21 | 212.9 | 0.33 | 18.79 |
| GBT | 246.56 | 0.27 | 20.94 | 270.80 | 0.26 | 19.85 | 337.60 | 0.34 | 22.78 | 266.84 | 0.24 | 21.36 | 278.94 | 0.27 | **20.94** |
| Ours (Geom Prior) | 172.42 | 0.32 | 21.11 | 204.66 | 0.33 | **21.18** | 252.78 | 0.37 | **23.98** | 224.73 | 0.26 | 21.21 | 219.64 | 0.36 | 20.66 |
| Ours | **61.9** | **0.16** | **22.30** | **98.5** | **0.19** | 20.53 | **167.9** | **0.24** | 23.65 | **82.7** | **0.15** | 22.08 | **122.2** | **0.24** | 20.19 |

# C   Ablation Study for Geometry Model

We further evaluate novel view image estimates from the stand-alone geometry model alongside other baselines. From Tables 5, 6, 7, we observe the geometry model outperforming the full version of our method in terms of PSNR in many cases. This is a consequence of the geometry model often rendering blurry coarse images as the view deviates from the context images. As PSNR, being a measurement favoring lower mean squared error across pixels, encourages mean pixel color, blurry images tend to score higher. However, significant drops in FID and LPIPS indicate lower image quality and perceptual dissimilarity, highlighting the importance of the 2D prior module in imagining missing details.

Table 7: Quantitative evaluation metrics for object-centric scene novel view synthesis on the hydrant category from CO3D.

| | 1 View | | | 2 Views | | | 5 Views | | |
|---|---|---|---|---|---|---|---|---|---|
| | FID↓ | LPIPS↓ | PSNR↑ | FID↓ | LPIPS↓ | PSNR↑ | FID↓ | LPIPS↓ | PSNR↑ |
| PixelNeRF [69] | 343.89 | 0.75 | 13.31 | 319.96 | 0.74 | 13.94 | 286.30 | 0.71 | 14.59 |
| SparseFusion [76] | 272.72 | 0.81 | 13.05 | 255.05 | 0.78 | 13.55 | 231.73 | 0.71 | 14.91 |
| Ours (Geom Prior) | 322.46 | 0.70 | **14.53** | 279.16 | 0.65 | **15.37** | 216.34 | 0.57 | **17.16** |
| Ours | **75.63** | **0.59** | 13.02 | **73.47** | **0.56** | 13.48 | **70.62** | **0.54** | 14.15 |

# D   Evaluation Setup

For computing evaluation metrics, we select 10 objects per category and sample 32 uniformly spaced camera poses from the held-out test split. We then randomly select a specified number of context views from the camera poses and evaluate novel view synthesis results on the rest of the poses.

# E   Additional Training and Evaluation on ShapeNet

We additionally train and evaluate our method and baselines on the cars category of the ShapeNet [5] synthetic dataset of object renderings. We train all methods on 2458 training car objects each containing 50 views. We randomly sample 1 to 3 context views during training. For evaluation, we randomly pick 10 objects from the test set with 251 views per object. We randomly sample one context view and evaluate on all other novel views. The final metrics are averaged across all views and objects. Quantitative results in Table 8 show improvements in FID and LPIPS over other baselines, indicating sharper image quality and diversity in synthesized results in comparison to the image distribution of ShapeNet cars and more faithful structure and texture of novel view reconstructions.

Table 8: Quantitative evaluation metrics for ShapeNet [5] cars category. For all baselines, only a single context view was provided as input to compute the metrics over novel views.

|  | FID↓ | LPIPS↓ | PSNR↑ |
|---|---|---|---|
| PixelNeRF [69] | 154.96 | 0.14 | **23.32** |
| SparseFusion [76] | 127.61 | 0.19 | 19.82 |
| Ours | **101.86** | **0.13** | 20.29 |

# F   Additional Qualitative Results

## F.1   Object-Centric Scenes

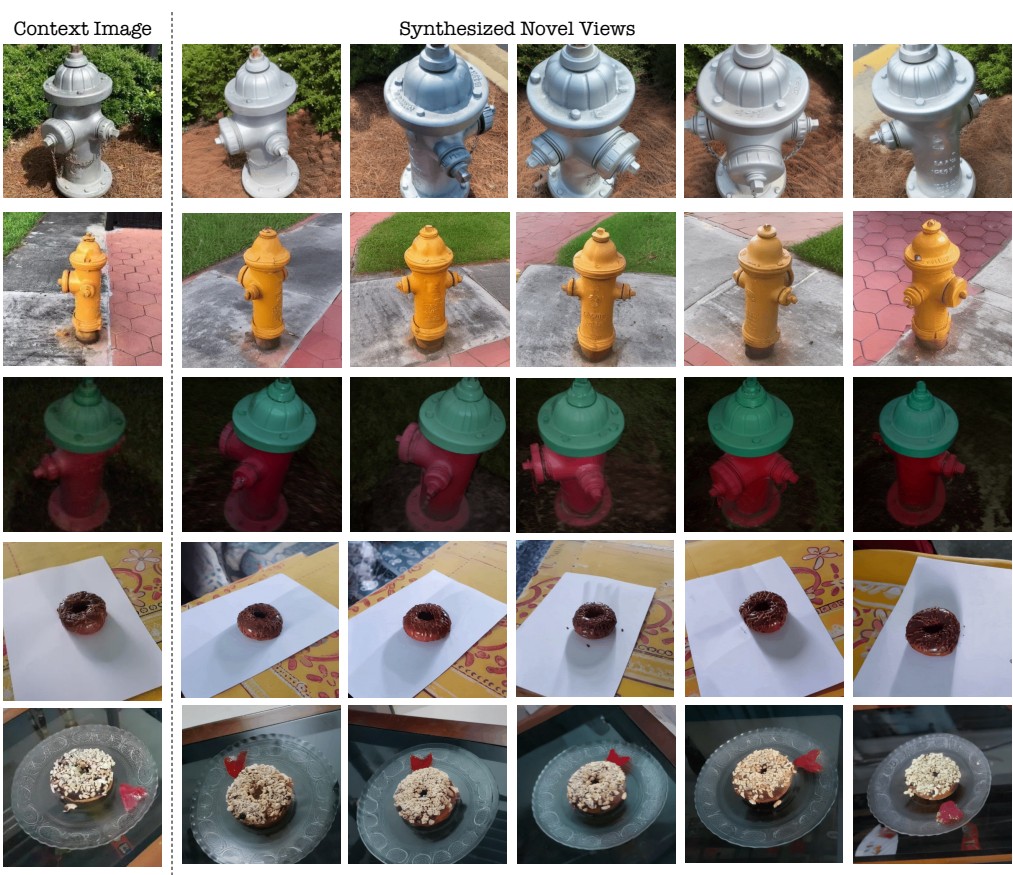

## F.2 Open-Set Categories

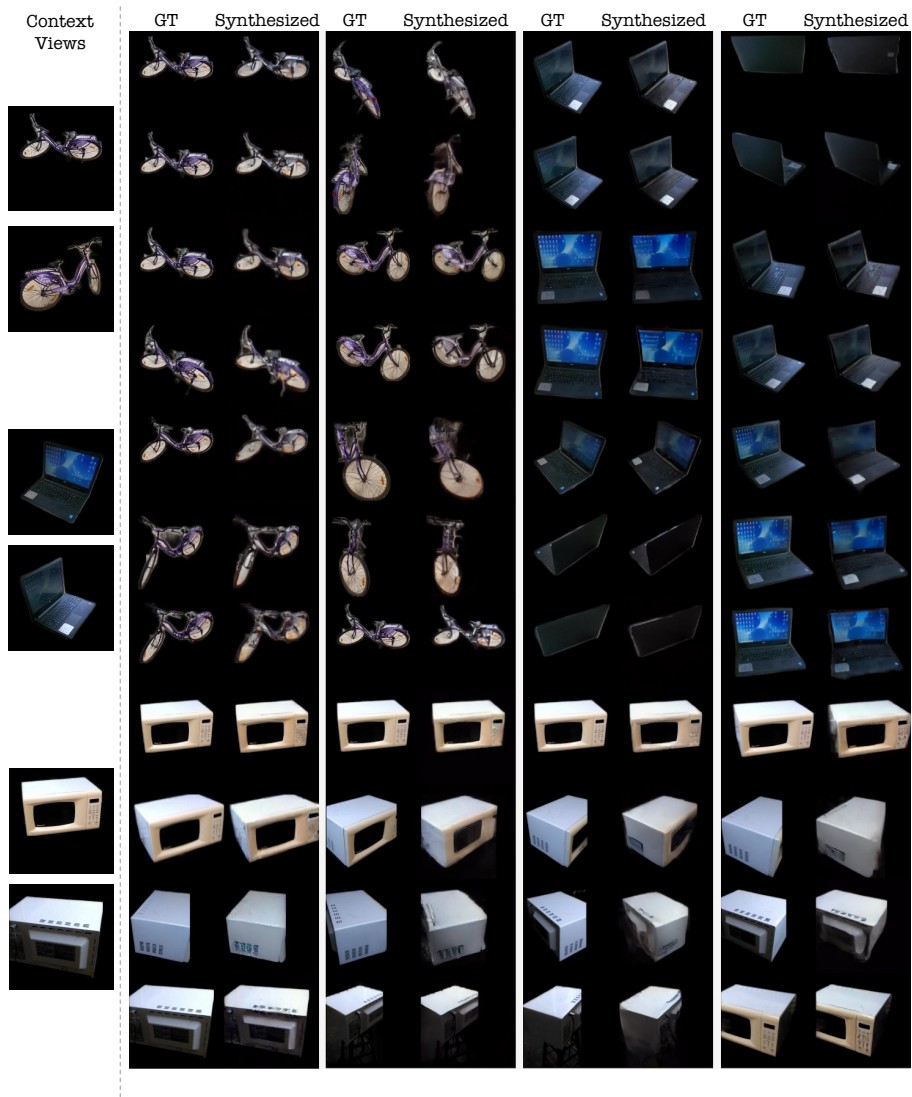

