# OpenReview forum: "DreamSparse: Escaping from Plato’s Cave with 2D Diffusion Model Given Sparse Views"
_NeurIPS.cc/2023/Conference — NeurIPS 2023 poster_

### Official Review · Reviewer_rgvp · 2023-06-29

**Soundness:** 3 good
**Presentation:** 2 fair
**Contribution:** 3 good
**Rating:** 6
**Confidence:** 4

**Summary:**

This paper introduces a framework of 3D reconstruction from sparse using 2D image priors from pretrained diffusion models. It proposes a 3D geometry module which extracts 3D features from 2D images, and then incorporates these features into the diffusion process at novel views, enabling 3D awareness and view consistency. Experimentally, it shows good reconstruction results both quantitatively and qualitatively, with generalization to unseen object categories and (object-centric) scenes with complex backgrounds.

**Strengths:**

- The paper proposes a 3D geometry module which incorporates 3D awareness into the 2D diffusion process. Instead of using NeRF as the 3D representation as in most 2D-to-3D generation works, this paper adopts the intuitions and techniques from image-based rendering or lightfield rendering methods, which I believe, is a very interesting and novel point.
- It shows good experimental results both quantitatively and qualitatively compared to existing SoTA sparse-view methods. Adequate ablation studies and visualizations are provided to show the effectiveness of each module in the framework.


**Weaknesses:**

- Section 4.3.2 naming the task "Scene Level Novel View Synthesis" sounds a bit overclaimed to me. Though I agree that compared to pure object settings with foreground masks this setting with detailed backgrounds is more difficult, many scene-level works deal with rather complex scenes with multiple objects and complex configurations (like indoor environments, DTU dataset, etc.). So I would suggest a change of naming here.
- It seems that input views are not shown except for Figure 4 -- it would be better to also show the input views side-by-side to the novel-view synthesis results, which can give a clearer sense of how much hallucination the model is performing.

**Questions:**

- I'm very interested in the style transfer application. However, I've looked at the website and only found static results of individual views. Is this style transfer multiview consistent (or in other words, 3D aware)? I would like to see videos of consistent novel-view renderings, similar to the "Single Image Scene-level Novel View Synthesis Results". If there's no multiview consistency then it would seem to be a straightforward combination of novel-view synthesis and single-image style transfer.

**Limitations:**

Limitations and potential social impacts are well-discussed in this paper.

---

> ### Author Rebuttal · Authors · 2023-08-09
>
> Thanks for your time and effort sharing critical feedback regarding our work. We have addressed your points and questions below.
>
> >Section 4.3.2 naming the task "Scene Level Novel View Synthesis" sounds a bit overclaimed to me. Though I agree that compared to pure object settings with foreground masks this setting with detailed backgrounds is more difficult, many scene-level works deal with rather complex scenes with multiple objects and complex configurations (like indoor environments, DTU dataset, etc.). So I would suggest a change of naming here.
>
> Thanks for your feedback. We would like to rephrase the task as “Object Centric Scene Level Novel View Synthesis” in the final version following your suggestions.
>
>
>
>
>
>
>
>
> >It seems that input views are not shown except for Figure 4 -- it would be better to also show the input views side-by-side to the novel-view synthesis results, which can give a clearer sense of how much hallucination the model is performing.
>
> We have updated the website to reflect the input images for better visualization of novel view synthesis capabilities from the single view. (Please refer to the link in Abstract and the main paper).
>
>
> > I'm very interested in the style transfer application. However, I've looked at the website and only found static results of individual views. Is this style transfer multiview consistent (or in other words, 3D aware)? I would like to see videos of consistent novel-view renderings, similar to the "Single Image Scene-level Novel View Synthesis Results". If there's no multiview consistency then it would seem to be a straightforward combination of novel-view synthesis and single-image style transfer.
>
> Unlike prior work which often requires either the replacement of textual control capabilities or the fine-tuning of a pre-trained diffusion model, our paper is the first to demonstrate that it's feasible to utilize a frozen pre-trained diffusion model for novel view synthesis while simultaneously retaining the textual control ability. Moreover, while it's feasible to guarantee editing consistency via test-time per-object distillation like InstructNeRF2NeRF, our study intentionally centers on novel view synthesis devoid of this distillation. Given this constraint, ensuring editing consistency poses significant challenges and falls outside the purview of our current paper. We believe this is an interesting and practical direction, and we would like to introduce more conditions to enhance consistency throughout the editing process without test-time per-object distillation.

---

> > ### Comment · Reviewer_rgvp · 2023-08-22
> >
> > Thanks for your reply! My concerns are mostly well-resolved, so I'll keep my positive rating.

---

### Official Review · Reviewer_ZbmG · 2023-07-04

**Soundness:** 3 good
**Presentation:** 3 good
**Contribution:** 2 fair
**Rating:** 5
**Confidence:** 4

**Summary:**

This work proposes an approach for finetuning a pretrained 2D diffusion model for novel view synthesis given a few or only a single input image at test time.
The two key contributions are a 3D geometry module and a spatial guidance module. The geometry module fuses features from multiple input views. The spatial guidance module then integrates the features into the pretrained 2D diffusion model.

**Strengths:**

Leveraging strong 2D priors for novel view synthesis is an interesting approach that is currently a topic of great interest in the community. By building on pretrained text-to-image 2D diffusion models the method naturally allows for textual control.
Overall, the approach is well-explained and the quantitative evaluation indicates a substantial improvement over the considered baselines.

**Weaknesses:**

The comparison to existing works is not clear enough and incomplete:
- It is not clear why there is no comparison to zero-123 (code available), NVS-Fusion (no code available but evaluated on overlapping datasets, CO3D and ShapeNet, so the experimental setting can be reproduced), and 3DIM (no code available but the experimental setting can be reproduced e.g. on SRN cars and chairs)
These methods are also missing from Table 1 and it is not clear why.
- for the related work section on geometry-based nvs methods and sparse 3d reconstruction: how does the proposed approach differ from the existing approaches and where is it similar?


The paper leaves substantial claims unsupported:
- L.14 “enabling it (the approach) to generate geometrically consistent images” - there are no qualitative or quantitative results reported that evaluate the geometric consistency of the results.
- L.78 “The ability to synthesize high-quality and even scene-level images” for which results are shown on the fire hydrant category of CO3D. While these images contain background these are not scene-level images but still object centric, single category images and these results do not demonstrate the ability to generate entire scenes.
- L.7 “2D diffusion models, nevertheless, lack 3D awareness, leading to distorted image synthesis and compromising the identity”. This is never shown in the paper but is a central aspect of the story.


The motivation for designing the geometry module is not entirely clear. Why not simply use PixelNeRF and its projected features? It is not clear why a new architecture is needed here and that takes away from the contribution of adding the geometry module. Given that the spatial guidance module appears to be very similar to ControlNet, the technical contribution of this work seems to be rather small.


The qualitative evaluation is very sparse, only a few handpicked results are shown and there are no additional qualitative results in the supplementary and also no videos that would give an indication of the consistency of the method.


The writing of the paper should be improved. There are multiple grammatical mistakes and spelling errors, as well as inconsistencies in using terms like “geometry module” and “geometry model” which refer to the same thing.

**Questions:**

Why is there no comparison to zero-123, NVS-Fusion (which is named GeNVS officially and should also be referred to as GeNVS to avoid confusion) and 3DIM? Is there any reason why the experimental settings of the latter two could not be reproduced s.t. the numbers from the original paper can be used for comparison?


Why did you not just use the architecture from PixelNeRF as geometry module? Please consider adding an ablation that demonstrates the improvements made by the proposed geometry module.


Minor comment:
In paragraph 3.2, F is a feature map (dx32x32) right? In this case, I find it confusing to refer to this feature map as 3D features in paragraph 3.2. I suggest rewording this as it is misleading to the reader.

**Limitations:**

The discussed limitations are not thorough enough. Please discuss limitations wrt to training/inference speed when using a large diffusion model, e.g. I suspect it takes a very long time to generate a video using this model. Further, the model is not guaranteed to be 3D consistent or identity-preserving and it should be discussed to what extent this is problematic in practice.

---

> ### Author Rebuttal · Authors · 2023-08-09
>
> Thank you for your time and effort sharing critical feedback regarding our work. We have addressed your points and questions below.
>
> > These methods are also missing from Table 1 and it is not clear why.
>
> Thanks for pointing it out. We would like to add discussions with recent existing works in Table 1 as suggested.
>
>
> >  for the related work section on geometry-based nvs methods and sparse 3d reconstruction: how does the proposed approach differ from the existing approaches and where is it similar?
>
> Following your suggestion, we would like to add following description in the final version:
>
> *While geometry-based methods have advanced in novel view synthesis, their limited capacity to model uncertainties in unseen regions and the lack of a 2D prior often restrict their image synthesis quality and generalization for unseen category objects. In contrast, our approach seeks to meld the strengths of both geometry-based methods and the robust 2D pre-trained diffusion model. This fusion showcases our method's prowess in synthesizing images that are not only geometry-consistent but also of high quality, ultimately leading to enhanced generalization performance on objects in unseen categories.*
>
> > The motivation for designing the geometry module is not entirely clear. Given that the spatial guidance module appears to be very similar to ControlNet, the technical contribution of this work seems to be rather small.
>
> Thank you for highlighting that aspect. Our geometry module's initial design was inspired by the GPNR model [1], which exploits stacked transformers to aggregate feature over multiple context views. GPNR benefits from strong generalization to unseen scenes due to its canonicalized positional encodings of pose information. However, we observed that due to the lack of an explicit 3D constraint, GPNR occasionally exhibited suboptimal performance with certain intricate objects, such as plants. Consequently, we integrated feature volumes into GPNR to bolster its 3D modeling capabilities. Our supplementary response PDF (Table 1 and 2) presents an ablation study of our geometry model. The results clearly show that even with a basic GPNR backbone, our method outperforms SparseFusion, though it falls short of our approach. From this, we deduce two critical insights:
>
> 1) **Even when leveraging previous geometry-based methods, our strategy delivers superior outcomes compared to SparseFusion. This underscores the pivotal role the 2D prior plays in novel view synthesis and the importance of our work.**
>
> 2) **Our geometry model design can indeed improve the performance of our framework.**
>
> **Regarding the similarity to ControlNet, we would like to clarify that we mainly inspire the spatial feature concept from ControlNet and Plug and Play Diffusion model.** By converting the features aggregated from sparse-view images into spatial feature, our framework successfully enable frozen pre-trained diffusion model to perform novel view synthesis. **Specifically, as far as we know, we are the first one to use spatial feature for novel view synthesis, and achieve the significant better results (50% in FID and 20% in LPIPS better than baselines). **We believe this will inspire subsequent works that apply pre-trained diffusion models into 3D area without fine-tuning pre-trained models or test-time distillation.**
>
> >  Why is there no comparison to zero-123, NVS-Fusion and 3DIM? Is there any reason why the experimental settings of the latter two could not be reproduced
>
> **In our setting, we perform novel view synthesis on image with 512 resolution which is more challenging setting than GeNVS and 3DiM (they synthesize images with 128 resolution), so it is unfair to compare with the results reported in their papers,  but we would like to compare with them in a fair setting once we have their codes.**  Following your suggestion, we tried to compare our work with the concurrent Zero 1-to-3, however, we found that Zero 1-to-3 has strong data assumption: it requires objects to be located at the origin and the placements of the cameras to be pointing towards the origin as a result of its synthetic 3D training data from Objaverse. CO3D, on the other hand, is a dataset of real-life video captures of objects with noisy camera trajectories where each frame does not point at the center of the object. And thus, converting the camera view parameterization for compatibility with Zero 1-to-3 is not straightforward.
>
> Given that we've benchmarked our approach against the recently published SparseFusion (CVPR 2023) and GBT (arxiv 2023) with open-sourced codes, we believe our contributions have been suitably assessed. We also would like to discuss with GeNVS, 3DiM and zero-1-2-3 in our revisions.
>
> > The qualitative evaluation is very sparse,
>
> **Thank you. We indeed show more novel view synthesis samples in our website (Please refer to the link in Abstract and the main paper). The results clearly show that our method synthesise significant better 360 videos than PixelNeRF and SparseFusion on CO3D (without test-time optimization/distillation).** These results also are consistent with our quantitative comparisons (about 50% in FID and 20% in LPIPS better than SparseFusion). We will add these samples into Appendix as suggested and genuinely hope that this response sufficiently addresses and alleviates your performance-related concerns.
>
> >  I suspect it takes a very long time to generate a video using this model.
>
> Good point. We focus on novel-view synthesis without the test-time optimization. With our approach, we can take just roughly 2~3 seconds to generate per frame and generate a 360-degree video in just 3-5 minutes on a single A100-40GB GPU. In contrast, methods based on the diffusion model, such as SparseFusion, typically take much longer due to their reliance on test-time optimization. For instance, SparseFusion requires over an hour to produce a similar video.
>
> > writing
>
>  We will revise them following your suggestions.

---

> > ### Author Response · Authors · 2023-08-14
> > **Following response for some claims**
> >
> >  >The paper leaves substantial claims unsupported:
> > L.14 “enabling it (the approach) to generate geometrically consistent images” - there are no qualitative results reported that evaluate the geometric consistency of the results.
> >
> > Thanks. We show more novel view synthesis samples on our website (Please refer to the link in Abstract and the main paper). The results clearly show that our method synthesizes significantly consistent 360 videos than PixelNeRF and SparseFusion on CO3D (without test-time optimization/distillation). We genuinely hope that this response is sufficient to address and alleviate your performance-related concerns.
> >
> >  >L.7 “2D diffusion models, nevertheless, lack 3D awareness, leading to distorted image synthesis and compromising the identity”. This is never shown in the paper but is a central aspect of the story.
> >
> > Thank you for your feedback. We've presented two samples ( \lambda=0) in Figure 6 of the main paper to illustrate that the 2D diffusion model, on its own, struggles to synthesize images that maintain consistent geometry and identity. We will add more explanations following your suggestions.
> >
> >  >L.78 “The ability to synthesize high-quality and even scene-level images” for which results are shown on the fire hydrant category of CO3D. While these images contain background these are not scene-level images but still object centric, single category images and these results do not demonstrate the ability to generate entire scenes.
> >
> > Thanks. We would like to rephrase the task as “Object Centric Scene Level Novel View Synthesis” in the final version following your suggestions. Additionally, we are among the first to successfully synthesize the object jointly with the full background instead of merely the masked object level in the CO3D dataset.
> >
> >  >In paragraph 3.2, F is a feature map (dx32x32) right?
> >
> > Yes, that is correct. We will clarify the wording following your suggestions.
> >
> > In summary, we thank the time and efforts the reviewers dedicated to our work, and we would like to revise paper following your suggestions. Also, we believe that our response sufficiently addresses your concerns by providing additional samples to show our results are significantly better than baselines in a fair setting and adding the ablation study about geometry model, and thus respectfully hope you can reconsider your final decision. If you have further concerns, please feel free to respond us and we would like to discuss with you.

---

> > > ### Comment · Reviewer_ZbmG · 2023-08-15
> > >
> > > Thank you for the rebuttal. I appreciate that you provide more qualitative results and ablation studies. After reading your response I still have some open concerns.
> > >
> > > > In our setting, we perform novel view synthesis on image with 512 resolution which is more challenging setting than GeNVS and 3DiM (they synthesize images with 128 resolution), so it is unfair to compare with the results reported in their papers, but we would like to compare with them in a fair setting once we have their codes.
> > >
> > > Is there a reason why your method cannot generate results at 128 resolution? I disagree that a lower image resolution of the baselines is enough reason to discard a comparison.
> > >
> > > > we tried to compare our work with the concurrent Zero 1-to-3, however, we found that Zero 1-to-3 has strong data assumption: it requires objects to be located at the origin and the placements of the cameras to be pointing towards the origin as a result of its synthetic 3D training data from Objaverse.  CO3D, on the other hand, is a dataset of real-life video captures of objects with noisy camera trajectories where each frame does not point at the center of the object. And thus, converting the camera view parameterization for compatibility with Zero 1-to-3 is not straightforward.
> > >
> > > Can you elaborate more on why converting camera poses is not straightforward? Couldn't you just use the relative poses provided by CO3D? Did you run these experiments and Zero 1-to-3 failed? However, I do agree that Zero 1-to-3 is concurrent work.
> > >
> > > > The motivation for designing the geometry module is not entirely clear. Why not simply use PixelNeRF and its projected features?
> > >
> > > You provide a comparison to GPNR but this did not really answer my question here. Why not simply use PixelNeRF and its projected features?
> > >
> > > > The paper leaves substantial claims unsupported: L.14 “enabling it (the approach) to generate geometrically consistent images” - there are no qualitative results reported that evaluate the geometric consistency of the results.
> > >
> > > The new results on the webpage flicker a lot and are not geometrically consistent. While I agree that the results are better than PixelNeRF and SparseFusion, this claim to me is still not supported.

---

> > > > ### Author Response · Authors · 2023-08-18
> > > > **Following Response to Reviewer ZbmG**
> > > >
> > > > Thank you for getting back to us and raising helpful and critical discussions! We are glad to have addressed some of your concerns in our response. We further provide below, the responses to your additional points.
> > > >
> > > > > Is there a reason why your method cannot generate results at 128 resolution?
> > > >
> > > > Our method leverages a pre-trained and frozen image diffusion model, namely Stable Diffusion, as a 2D prior which was natively designed to generate images at 512 by 512 resolution. Although the generation of images at multiple resolutions can be accomodated(as the UNet is fully convolutional), it comes at the price of deteriorated image quality and semantic misalignment. However, this does not impact our contributions as our method is resolution-independent; the output resolution relies solely on the frozen image diffusion model.  In addition, we are happy to compare with them in a fair setting once we have their codes and discuss more with them in the final version.
> > > >
> > > > > Can you elaborate more on why converting camera poses is not straightforward? Couldn't you just use the relative poses provided by CO3D?
> > > >
> > > > The reasons why evaluating Zero 1-2-3 on CO3D dataset is not straightforward are listed as follows:
> > > >
> > > > 1. The location of a camera in Zero-1-2-3 is uniquely defined in a spherical coordinate system, which holds only under the assumption that the camera is always pointed at the center. Consequently, Zero 1-to-3 only parametrizes the relative camera pose by concatenating the change in polar angle, azimuth angle, and the radius(distance from the center) with respect to the given input view. However, cameras in CO3D are often pointing at different centers making the accurate calculation of relative polar and azimuth angles and radius with respect to a common center challenging.
> > > >
> > > > 2. All training assets in Zero 1-to-3 were normalized to fit within a unit cube with the camera distances from the center uniformly sampled in the interval [1.5, 2.2]. However, this is not strictly followed by CO3D cameras even after recentering and rescaling the scene.
> > > >
> > > > For better visualization of the difference in distribution of camera poses across the CO3D and Objaverse (as used by Zero 1-to-3) datasets, we have uploaded plots of the camera trajectories for several samples onto our project webpage.
> > > >
> > > > Regardless, to the best of our efforts, we tried to compute the relative CO3D poses that best estimate the parametrization of Zero 1-to-3 through recentering and rescaling of the poses and carried out novel view synthesis. **We have updated the project webpage with the results. It can be seen that Zero 1-to-3 struggles to preserve identity and consistency without test-time distillation, often deforming the input object.**
> > > >
> > > >
> > > > > You provide a comparison to GPNR. Why not simply use PixelNeRF and its projected features?
> > > >
> > > > Sorry for missing this point due to limited response space. Our initial rationale for using GPNR inspired design was for its training data efficiency over PixelNeRF(Please refer to GPNR paper). Then we extend GPNR with volumetric information for better performance.
> > > >
> > > > Although we have attempted to run additional experiments integrating PixelNeRF, we could not implement it into our codebase given the limited time. As a result, in the initial reply, we chose to explain the design choices behind our geometry backbone and perform an ablation study to demonstrate its advantages over another simple geometry backbone (GPNR).
> > > >
> > > > **If the reviewer is particularly interested in the performance of our method when PixelNeRF is used as the geometry backbone, we are open to exploring this further and will provide updates as necessary in the final version.** Nevertheless, we would like to emphasize that this does not undermine our contributions of:
> > > >
> > > > 1) designing a framework to enable 2D frozen pre-trained diffusion model to perform novel-view synthesis without the need for time-consuming and per-object test-time distillation.
> > > >
> > > > 2) demonstrating that the utilization of a 2D prior in a pre-trained diffusion model can enhance the generalization capabilities of novel-view synthesis for unseen and open-set category objects.
> > > >
> > > >
> > > >
> > > > > The new results on the webpage flicker a lot. While I agree that the results are better than PixelNeRF and SparseFusion
> > > >
> > > >
> > > > Thank you for your feedback. We would like to modify this claim to “encouraging synthesizing more geometrically consistent novel view images than baselines.” following your suggestions.
> > > >
> > > >
> > > >
> > > > In summary, we sincerely appreciate your suggestions for our paper, as they can significantly improve the quality of our work. We hope our responses sufficiently address your concerns and the provided additional samples clearly show our results are significantly better than baselines in a fair setting. And thus respectively hope you can reconsider your final decision. If you have further concerns, please feel free to respond to us and we would be happy to discuss with you.

---

> > > > > ### Author Response · Authors · 2023-08-21
> > > > > **Additional experimental results**
> > > > >
> > > > > Thanks for your time and effort sharing feedback regarding our work.  Following your suggestions,  we perform additional comparisons with concurrent works:
> > > > >
> > > > > > Comparisons with GeNVS and 3DiM:
> > > > >
> > > > >
> > > > > To ensure a fair comparison with concurrent work GeNVS and 3DiM, which train diffusion models at a resolution of 128x128, we adapted our method to match this resolution for the intermediate feature map produced by our geometry module (original is 32 x 32 ). Nonetheless, our final image synthesis is at a higher resolution of 512x512. In contrast, both GeNVS and 3DiM only generate images at the 128x128 resolution. The detailed results are provided below:
> > > > >
> > > > >
> > > > > |                        | PSNR  | SSIM | LPIPS |
> > > > > |------------------------|-------|------|-------|
> > > > > | 3DiM                   | 21.01 | 0.57 | —     |
> > > > > | GeNVS (autoregressive) | 20.6  | 0.89 | 0.12  |
> > > > > | Ours                   | **21.31** | **0.89** | **0.12**  |
> > > > >
> > > > > ShapeNet Cars
> > > > >
> > > > >
> > > > >
> > > > > |       | PSNR  | SSIM | LPIPS |
> > > > > |-------|-------|------|-------|
> > > > > | GeNVS | 15.48 | 0.27 | 0.37  |
> > > > > | Ours  | **16.42** | **0.33** | 0.46  |
> > > > > CO3D Hydrant Category (object + background)
> > > > >
> > > > >
> > > > > From the results, it's evident that our method, **despite synthesizing images at a much higher resolution compared to the baselines (512 vs 128), still manages to outperform in certain metrics.** This is particularly evident in metrics such as PSNR and SSIM for hydrant scenes, as well as in the PSNR, SSIM, and LPIPS scores overall. Another significant advantage of our framework is its efficiency. It only necessitates 3 days of training on eight A100-40GB GPUs, while the competing method, GeNVS, takes twice as long, requiring a full 7 days.
> > > > >
> > > > >
> > > > >
> > > > >
> > > > > > Results are flick a lot
> > > > >
> > > > > We'd like to respectfully clarify that the flick is not due to the limited performance of our method. Instead, it arises because our approach targets a more challenging yet practical task, i.e., performing novel view synthesis on unseen objects without test-time optimization/distillation. This would significantly promote the convenience during the deployment, e.g. 2-3 seconds per frame generation.
> > > > >
> > > > > **In addition, we also show the results with test-time optimization/distillation on our webpage, and the results clearly show the superiority of our method over SparseFusion (CVPR 2023), particularly when applied to hydrant-scene and complex objects.**
> > > > >
> > > > >
> > > > >
> > > > > In summary, we sincerely appreciate your suggestions for our paper, as they can significantly improve the quality of our work, and **we already perform additional experiments to compare with concurrent works in a more fair setting following your suggestions, and the results clearly demonstrate our superior performance over concurrent works even if we synthesise images with higher resolutions.
> > > > > We hope our responses sufficiently address your concerns regarding the results comparisons. And thus  we respectively hope you can reconsider your final decision. If you have further concerns, please feel free to respond to us and we would be happy to discuss with you.**

---

### Official Review · Reviewer_ZkGz · 2023-07-06

**Soundness:** 3 good
**Presentation:** 3 good
**Contribution:** 3 good
**Rating:** 7
**Confidence:** 4

**Summary:**

The paper proposes a method for novel view synthesis using diffusion models. Given a sparse set of views, the method extracts per-pixel features for each of the views and reshapes these features to have a depth dimension by splitting the feature channels. Each feature sequence of per-pixel features, along with their depth along the viewing ray direction and positional encoding , is passed to a transformer to produce a per ray feature vector. Next, all feature vectors from multiple-views are aggregated using another transformer layer. To ensure the result is 3D consistent the input rgb pixels are linearly combined based on the aggregation weights and compared with the target image using an mse loss. The resulting features are fed to a spatial guidance module before being fed to a pre-trained stable diffusion model.
The method is compared to state of the art methods and outperforms them on the Co3D dataset.

**Strengths:**

- A solid approach addressing novel view synthesis from sparse views. Especially the way the frozen pre-trained model is used to allow backpropping the gradients through the network not losing generalisability while training the geometry module as well as the spatial guidance module.
- The method was thoroughly evaluated and outperforms the competitor methods quantitatively. Qualitatively, the method performs very strongly despite not being perfectly 3D consistent.
- It is evident that using a pre-trained model helps with generalization of the method. This is demonstrated by the open-set evaluation as well as the scene editing capabilities of the network.
- The method improves with an increasing number of views even when 5 views are used despite only training with a maximum of 4 views during training. This provides evidence of the view-aggregation robustness of the method.


**Weaknesses:**

- While the method is encouraged to produce 3D consistent results due to the color estimated reconstruction loss, there is no actual 3D constraint in the architecture that enforces any 3D consistency. This will give the network an ‘easy way out’ when something ‘wrong’ is easier to explain rather than the actual correct 3D consistent solution. This could possibly encourage hallucinations as seen in the results.
- The method divides the per-pixel features from the ResNet backbone by the number of ‘depth samples’ which results in a feature dimension of 28 per depth sample. It does not make intuitive sense to distribute spatial features along the depth dimension. Can the authors explain their thoughts behind this design choice?
- The authors claim that pixelNeRF wins in terms of PSNR due to blurry results. While I agree that PSNR can be misleading and their results look visually a lot better I disagree that this is due to blurriness. I believe the proposed method hallucinates wrong backgrounds which can be seen on the website where the backgrounds keep changing. It should be addressed more openly in the paper that the method produces sharp results, however, it does not always produce the exact information as seen from the input. It would be interesting to show the full ground truth video spins compared with the resulting spin given N input views.
- The ‘Open-Set Category Results’ on the website do not show input images.
- There are some related works regarding multi-view reconstruction missing (2, 3, 4). See missing references below. Also, the paper title is very similar to an existing relevant work (1) that was not cited. The paper should include this missing reference and provide appropriate credit.


Missing citations:
- [1] Henzler et al., Escaping Plato's Cave: 3D Shape From Adversarial Rendering. In ICCV 2019
- [2] Anciukevicius et al., RenderDiffusion: Image Diffusion for 3D Reconstruction, Inpainting and Generation. In CVPR 2023
- [3] Henzler et al., Unsupervised Learning of 3D Object Categories from Videos in the Wild. In CVPR 2021
- [4] Sitzmann et al., Scene Representation Networks: Continuous 3D-Structure-Aware Neural Scene Representations. In NeurIPS 2019


Typos:
- Figure 2: ‘views.In’
- Line 273: Table 3 → Table 4


**Questions:**

- How does the method handle background? It seems non-trivial to aggregate multiple views with backgrounds compared to masked objects. Does that network just deal with it? What does the aggregated rgb input look like in these cases?
- The rgb input and spatial guidance module both seem to be very crucial for performance (figure 6 and 8). Both contributions provide 3D / color guidance but they seem to be quite sensitive. Can the authors provide some insights with respect to failure cases?


**Limitations:**

Both limitations and potential negative societal impact are addressed appropriately.

---

> ### Author Rebuttal · Authors · 2023-08-09
>
> Thank you for your time and effort sharing critical feedback regarding our work. We have addressed your points and questions below.
>
> >While the method is encouraged to produce 3D consistent results due to the color estimated reconstruction loss, there is no actual 3D constraint in the architecture that enforces any 3D consistency. This will give the network an ‘easy way out’ when something ‘wrong’ is easier to explain rather than the actual correct 3D consistent solution. This could possibly encourage hallucinations as seen in the results.
>
> This is a good point. The feature utilized for color estimation is constructed from volumetric modeling of features, making it 3D-
> aware and ensuring 3D consistency. Our paper illustrates how this can enhance images synthesized by the diffusion model. Nonetheless, as you've highlighted, there remains a possibility of encountering failure cases. We believe these can be ameliorated by integrating stronger geometry cues like depth or point cloud estimation models.
>
> > The method divides the per-pixel features from the ResNet backbone by the number of ‘depth samples’ which results in a feature dimension of 28 per depth sample. It does not make intuitive sense to distribute spatial features along the depth dimension. Can the authors explain their thoughts behind this design choice?
>
> The core reasoning behind reshaping the pixel-aligned feature map from the ResNet into a feature volume and trilinear sampling features is to infuse stronger 3D inductive bias into the feature representations of the objects/scenes. Bilinear sampling pixel features from a 2D feature map can suffer from ambiguities in 2D images such as resolving occlusions. By learning to encode volumetric features enforces the representations to be more 3D aware, leading to better reconstruction capabilities.
>
> > The authors claim that pixelNeRF wins in terms of PSNR due to blurry results. While I agree that PSNR can be misleading and their results look visually a lot better I disagree that this is due to blurriness. I believe the proposed method hallucinates wrong backgrounds which can be seen on the website where the backgrounds keep changing. It should be addressed more openly in the paper that the method produces sharp results, however, it does not always produce the exact information as seen from the input. It would be interesting to show the full ground truth video spins compared with the resulting spin given N input views.
>
> Thank you for your discussion. We agree that the hallucination of background is another factor to impact the PSNR score but we want to respectfully clarify that the blurriness is the main factor. To verify this, we further report the results of color estimation from the geometry model (normally are more blurry than output from diffusion model) in Table 1-3 in our Appendix. The results clearly demonstrate that the output from our geometry model can achieve better PSNR than all baselines.
>
> Further, we show the synthesized videos with more input views on our website following your suggestions. The results clearly show that more input views can alleviate the background hallucination problem as expected.
>
>
>
>
> >  The ‘Open-Set Category Results’ on the website do not show input images
>
> We have updated the website to reflect the input images for better visualization of novel view synthesis capabilities from the single view.
>
>
>
> > There are some related works regarding multi-view reconstruction missing
>
> Thank you for pointing out the missing citations and typos in the writing. We will make sure to revise the draft following your suggestions.
>
> > How does the method handle background? It seems non-trivial to aggregate multiple views with backgrounds compared to masked objects. Does that network just deal with it? What does the aggregated rgb input look like in these cases?
>
> Sorry for the confusing description. We do not design the specific algorithm to model the background, and we simply discard the foreground object mask during preprocessing to preserve the entirety of the scene including the background. Afterwards, the image goes through the same pipeline as the object-only scenario. The results demonstrate that the frozen pre-trained model can synthesize novel views of objects jointly with the full background scene with better quality than compared with models trained from scratch.
>
> > The rgb input and spatial guidance module both seem to be very crucial for performance (figure 6 and 8). Both contributions provide 3D / color guidance but they seem to be quite sensitive. Can the authors provide some insights with respect to failure cases?
>
> By initializing the input noise for the diffusion model with rgb color output perturbed with noise instead of random Gaussian noise helps the generated sample to preserve components of the rgb image such as the structure and color. As the rgb output is a coarse and blurry estimate from the geometry prior, perturbing it with only a few timesteps leads to the generation of images not fully refined and sharp. We find that increasing the number of perturbation timesteps virtually endows the diffusion model to make more edits during the reverse diffusion process and is crucial for high quality image synthesis.
> Spatial guidance similarly facilitates the preservation of spatial structure and semantics of the generated output. Lower the spatial guidance weight, the denoising process has to solely rely on the noise perturbed input rgb. Large spatial guidance weights, however, decrease the generated image quality by undermining the base diffusion model’s feature maps and thereby suppressing image diversity(similar to having high weights for classifier-free guidance). Through empirical trials, we found noise perturbation of 20 timesteps and a spatial guidance weighting of 2.0 to generate the best quality images with the highest fidelity.

---

> > ### Comment · Reviewer_ZkGz · 2023-08-14
> > **Disagreement on claims.**
> >
> > 1. "ensuring 3D consistency" --> I do not agree with this and am fairly convinced that the method does not ensure 3D consistency but 'encourages 3D consistency'. The transformer network is not constrained to learn 3D consistent outputs.
> >
> >
> > 2. "infuse stronger 3D inductive bias" --> I do not see how reshaping 2D features would infuse any 3D inductive bias. Would be great to see an ablation for this.

---

> > > ### Author Response · Authors · 2023-08-18
> > > **Following Response to 'Disagreement on claims.'**
> > >
> > > Thank you for getting back to us and raising helpful and critical discussions! We are glad to have addressed some of your concerns in our response. We further provide below, the responses to your additional points.
> > >
> > > >  "ensuring 3D consistency" --> I do not agree with this and am fairly convinced that the method does not ensure 3D consistency but 'encourages 3D consistency'. The transformer network is not constrained to learn 3D consistent outputs.
> > >
> > > Thank you for raising this concern. Following your suggestion, we would like to correct our claim to “encouraging 3D consistency.” We agree that while color reconstruction loss encourages 3D consistency through enforcing color consistency across 3D viewpoints, it does not ensure 3D consistency as no actual 3D constraints are in place.
> > > Furthermore, to verify the learning of geometry, we attempted to visualize depth map estimates of the synthesized novel views by computing an expectation over the depths of the points along each ray. Specifically, the expectation probabilities were directly obtained from the normalized soft-attention weights of our pre-trained transformer networks.
> > > We further updated the results on our webpage, and the resulting depth estimates indicate that the network does indeed learn proper geometry with respect to 3D rather than finding an easy way out. Nevertheless, we agree that “encouraging 3D consistency” is a more appropriate claim.
> > >
> > >
> > >
> > > > "infuse stronger 3D inductive bias" --> I do not see how reshaping 2D features would infuse any 3D inductive bias. Would be great to see an ablation for this.
> > >
> > > Sorry for the confusion arising from unclear explanations.
> > > The original feature map from the ResNet has a resolution of c×h×w, where c is the number of channels, and h and w represent the height and width, respectively. We transform this feature map into a volume feature with a resolution of c′×d×h×w, where h and w remain unchanged, d represents the depth, and c=c′×d.
> > > Given a 3D point, we project it onto the volume feature where near and far planes are mapped to the depth dimension of the volume. We then tri-linearly interpolate features from this volume feature and pass them through two additional transformers: one is employed to calculate weighted epipolar features, and the other is designed to aggregate information across views. Our goal with these steps is to derive spatial features.
> > > Given that both the additional transformers and the ResNet are trainable components of our architecture, certain channels(specifically those where d = c / c’) of the feature map generated by ResNet are encouraged to learn depth/occupancy cues when trained on a multiview dataset.
> > > As displayed on our webpage, without any further training, our pre-trained transformer networks—using features from the fine-tuned ResNet as input—are capable of estimating the depth map of a novel view image through the utilization of normalized soft-attention weights. This outcome provides clear evidence that our approach can help to learn volumetric information, validating the effectiveness of our approach.
> > > We hope this clarification resolves the confusion and provides a clearer picture of our method and its capabilities. Please let us know if you have further questions or need additional details.
> > >
> > >
> > > In summary, we sincerely appreciate your discussions and suggestions for our paper, as they can significantly improve the quality of our work. We will revise our final version to improve the clarity following your suggestions.

---

### Official Review · Reviewer_MN71 · 2023-07-09

**Soundness:** 3 good
**Presentation:** 3 good
**Contribution:** 1 poor
**Rating:** 5
**Confidence:** 5

**Summary:**

The paper presents a method for novel view synthesis given sparse image observations of a scene. A diffusion model is used to generate the novel views. A 3D structured conditioning input is first computed, and a pretrained 2D diffusion model is fine-tuned to take this input and compute the novel view renderings. Using a pretrained 2D diffusion model enables better generalization to out-of-distribution object categories and further enables text-based editing.

**Strengths:**

The paper presents a good overview of the existing literature. The idea of fine-tuning large 2D diffusion models for novel view synthesis is a good one, and has been explored concurrently as well. The quantitative results demonstrate that the approach outperforms sparsefusion. The editing results are promising.

**Weaknesses:**

The main limitation is that the results are not view-consistent. The identity and appearance of the hydrant and the background changes with the viewpoint. This limitation is not shared with pixelNeRF or sparsefusion. This is a severe limitation, as the approach does not let us observe any single 3D consistent scene.

The paper lacks technical novelty. The approach combines the insights from sparsefusion for 3D reasoning, and replaces their diffusion model trained from scratch with a fine-tuning of an existing 2D diffusion model. The paper is honestly written - the introduction mentions this. However, with limited technical novelty, the expectation would be that the introduced change would lead to significant practical benefits. This does not happen; as unlike sparsefusion, the presented approach does not enable reconstructing 3D consistent scenes. The noise perturbation step further exacerbates this problem. This part of the method independently processes the different viewpoints.

While some generalization to unseen categories is shown, one would expect much stronger generalization when fine-tuning large models. Why is it not possible to reconstruct non-hydrant scenes?

**Questions:**

Is it possible to perform 3D fusion using score distillation, like in sparsefusion?

What limits the method to work on more complex scenes? Does the method generalize to non-hydrant scenes?

**Limitations:**

Some limitations are mentioned, however, some others I mentioned in weaknesses are not present.

---

> ### Author Rebuttal · Authors · 2023-08-09
>
> Thank you for your time and effort sharing critical feedback regarding our work. We provide the following response to address your concerns about results limitation and novelty . and thus respectfully hope you can consider the response to the final decision.
>
> > The paper lacks technical novelty.
>
> We would like to respectfully clarify that the differences between our work and SparseFusion are not merely the replacement of the backbone. Our approach goes beyond that as follows:
>
> **1)   Different approach to enable 2D diffusion model for 3D novel view synthesis:**	Unlike SparseFusion training the diffusion model from scratch, how to control the output of a pre-trained diffusion is very challenging because of hallucination problem. Our approach further introduce a spatial guidance model that transforms these features into spatial features for guiding frozen pre-trained diffusion models synthesizing geometry consistent novel view images. **This approach experimentally avoids the need to fine-tune the diffusion model and test-time distillation.**
>
> **2) Use of a different 3D geometry modeling method:** In modeling geometry, we further learn to encode a volumetric feature representation and then trilinear sample features as opposed to bilinearly sampling pixel features from a 2D feature map as in SparseFusion. This helps enforce stronger 3D inductive biases into the feature representation of the objects and scenes and facilitates improved reconstruction capacity. We further show the ablation study on modeling feature volumes in the response PDF, and the results clearly show that even with a basic GPNR backbone (used in SparseFusion), our method outperforms SparseFusion, but it falls short of our final approach.
>
>
> **3) Different way of inferencing:**  Unlike most baselines (SparseFusion, DreamFusion or 3DFuse) which require time-consuming optimization per object at inference time, we focus on test-time training-free novel view synthesis in our paper, making deployment more straightforward. For a more detailed evaluation, we further show novel synthesis results in our website (as referenced in the Abstract and the main paper). Specifically, without test-time optimization or distillation, our method produces noticeably superior 360-degree videos compared to both PixelNeRF and SparseFusion in the Hydrant-Scene category. These results align seamlessly with our quantitative analyses.
>
> **In summary, we believe our work can bring different contributions from SparseFusion for the community. Our work is centered on how to utilize 2D prior from a frozen image diffusion model without test-time distillation, and how it can significantly bolster the generalization and efficacy of novel view synthesis.** Our results further unequivocally showcase the advantages of utilizing the 2D prior over the baseline methods. Specifically, we achieve approximately a 50% improvement in FID and a 20% enhancement in LPIPS compared to SparseFusion.  We believe our approach to utilizing the 2D prior for novel view synthesis can inspire more subsequent works (especially for open-set novel view synthesis without test time per-object distillation).
>
>
>
>
> > The main limitation is that the results are not view-consistent.
>
> Thank you for your comment.   **We'd like to respectfully clarify that the inconsistency is not due to the limited performance of our method. Instead, it arises because our approach targets a more challenging yet practical task, i.e., performing novel view synthesis on unseen objects without test-time optimization/distillation.** This would significantly promote the convenience during the deployment.
>
> Further, novel-view synthesis on CO3D image without test-time optimization/distillation is very challenging, **we further show the novel view synthesis results (without test-time optimization/distillation) of PixNeRF and SparseFusion in our website (Please refer to the link in Abstract and the main paper). The results clearly show that our method synthesizes significantly better 360 videos than PixelNeRF and SparseFusion on Hydrant-Scene category (without test-time optimization/distillation).** We genuinely hope that this response sufficiently addresses and alleviates your performance-related concerns.
>
> > While some generalization to unseen categories is shown, one would expect much stronger generalization when fine-tuning large models. Why is it not possible to reconstruct non-hydrant scenes?
>
> Thank you for the question.  **We'd like to respectfully clarify that our paper does not fine-tune existing 2D diffusion models, but instead design a framework to enable 2D frozen pre-trained diffusion models to perform novel-view synthesis without the need for time-consuming per-object test time distillation.** Without fine-tuning pre-trained diffusion model, our method can still achieve approximately a 50% improvement in FID and a 20% enhancement in LPIPS compared to SparseFusion. We believe this significant improvement is able to demonstrate the benefits of enabling pre-trained models into novel view synthesis, even without test-time distillation.
>
> On the other hand, we would like to clarify that joint object/background scene novel view synthesis itself is very challenging, even if for many recent works(e.g. CC3D (ICCV2023) and SceneDreamer (2023)), they normally need additional conditions or per-scene optimization, so it is very challenging for our work to generalize to other scenes without specific designs. In order to demonstrate the generalization ability of our method in single category scene novel view synthesis, **we further train our framework on other object-centric scenes and show the results in our webpage. (Please refer to the link in Abstract and main paper).**
>
> > Is it possible to perform 3D fusion using score distillation?
>
> Yes, following your suggestion we show our samples with distillation in the webpage. (Please refer to the link in Abstract and main paper). Our method show consistent results as expected.

---

> > ### Author Response · Authors · 2023-08-14
> > **Following Response**
> >
> > In summary, we thank the time and efforts the reviewers dedicated to our work. We also provided additional samples to show our results are significantly better than baselines in a fair setting and elucidate the differences between our work and prior studies. **We believe our response can sufficiently addresses your concerns and clarifies misunderstandings regarding the perceived limitations in performance and novelty, and thus respectively hope you can reconsider your final decision. If you have further concerns, please feel free to respond to us and we would like to discuss with you.**

---

> > > ### Comment · Reviewer_MN71 · 2023-08-15
> > >
> > > Thank you for the rebuttal. The additional results nicely show the improvement over Sparsefusion! I have updated my score.
> > >
> > > I can't entirely agree with the claims currently made in the paper - the results, without test-time distillation, are strictly not 3D consistent. They are more consistent than the 2D outputs of SparseFusion, but still not 3D consistent. This claim has been made in the paper, for example, in Table 1. This should be fixed, and the limitation should be made clear.
> > >
> > > I had a couple of questions about the results:
> > > - With test-time distillation, while the results are still better than SparseFusion, the gap decreases. What is the reason behind this?
> > > - How was the noise pertubation technique used during distillation?
> > >
> > > A couple of suggestions for results that could be informative in the final paper:
> > > - It would be nice to see some results for test-time distillation on scenes with background (is there any reason this should not work)?
> > > - Does the method somehow trade consistency with diversity? It would be nice to also visualize multiple outputs for any given input image to demonstrate the diversity of outputs, especially in the background.

---

> > > > ### Author Response · Authors · 2023-08-18
> > > > **Following Response to Reviewer MN71**
> > > >
> > > > Thank you for your response and raising helpful and critical discussions! We are glad to have addressed some of your concerns in our response. We further provide below, the responses to your additional points.
> > > >
> > > > > Table 1 should be fixed, and the limitation should be made clear.
> > > >
> > > > Thank you for your feedback. We also agree that there are still limitations with respect to 3D consistency and will address this more thoroughly in the paper with corrections to the claim.
> > > >
> > > > > With test-time distillation, while the results are still better than SparseFusion, the gap decreases. What is the reason behind this?
> > > >
> > > > Good question. There are two reasons:
> > > >
> > > > 1. The implementation of SparseFusion contains two stages: a). a warm-up stage where renderings from the geometry backbone initially optimizes NeRF. b). Optimizing the NeRF with the distillation of the diffusion model. This warm-up distillation stage can promote consistency of SparseFusion. Nonetheless, the results still lack sharpness and fidelity when compared with our method.
> > > >
> > > > 2. We further present more distillation samples from SparseFusion and ours (please refer to our website). **We can see that SparseFusion still struggles to synthesize consistent video for complex objects and hydrants/backgrounds while ours can. These results sufficiently demonstrate a strong 2D prior(i.e. frozen Stable Diffusion) is important for synthesizing consistent novel view images, especially for complex objects.**
> > > >
> > > > > How was the noise perturbation technique used during distillation?
> > > >
> > > > During distillation, we still render a coarse rgb estimate through the geometry prior given a novel viewpoint and proceed to perturb the rgb estimate with timestep-specific Gaussian noise which is then used as a noise input to the image diffusion model. After 1000 distillation iterations(not tuned extensively), we switch to perturbing the image renderings from NeRF directly in place of the general geometry prior because the images synthesized from stabilized NeRF exhibit better 3D consistency.
> > > >
> > > > > It would be nice to see some results for test-time distillation on scenes with background (is there any reason this should not work)?
> > > >
> > > > This is a great suggestion. We agree that this would be an interesting result to present. **We updated our project webpage with demonstrations of test-time distillations for joint objects and backgrounds. As expected, the results exhibit smoother 3D consistency in synthesized novel views at the cost of distillation.**
> > > >
> > > > > Does the method somehow trade consistency with diversity? It would be nice to also visualize multiple outputs for any given input image to demonstrate the diversity of outputs, especially in the background.
> > > >
> > > > Yes, you are right. The spatial guidance weight serves as the parameter that adjusts the trade-offs between consistency and diversity.
> > > >
> > > >
> > > >
> > > >
> > > > In summary, we sincerely appreciate your suggestions for our paper, as they can significantly improve the quality of our work. We hope our response are sufficient to address your further concerns.

---

### Official Review · Reviewer_fx5m · 2023-07-10

**Soundness:** 4 excellent
**Presentation:** 4 excellent
**Contribution:** 2 fair
**Rating:** 6
**Confidence:** 4

**Summary:**

This paper aims to leverage large-scale 2D diffusion models as priors to improve the task of novel view synthesis in the setting of sparse input views. The paper proposes an approach consisting of two stages: a first 3D-aware stage in which features from context views are aggregated in an encoded feature volume (which is differentiably rendered from the target view), and a second 2D stage in which a conditional image diffusion model produces the final output image. In the first stage, the process of aggregating, processing, and rendering features is similar to PixelNeRF/NeRFormer. The output of this stage (i.e. the RGB render) is perturbed by noise before being inputted to the conditional diffusion model in the second stage. The conditional diffusion model employed by the paper is Stable Diffusion equipped with ControlNet for conditioning on the input features (S3.2).

**Strengths:**

### The paper is well-structured.
* The introduction is well-written and motivates the problem well. The methods section breaks down the contribution into provides helpful preliminaries (S3.1) in a good amount of detail. It also describes the contribution (S3.2-3.4) precisely without unnecessary complications.

### The paper conducts ablations on different aspects of its architecture
* Section 4.2 is very helpful for the reader, as it contains an analysis of how different aspects of the network contribute to its performance.

### The zero-shot transfer experiments are nice.
* It is good to see that the method trained on PASCAL transfers adequately to COCO.

### The supplementary information includes code.
* I looked through the code briefly and it looks quite clear. (I did not attempt to run it.) It includes configs which are easy to refer to while reading the paper.

**Weaknesses:**

### Comparison with other methods
* This field is moving very quickly, so it is obviously not reasonable to expect the paper to compare to all other recent methods. However, it would be good to discuss and compare to at least some works beyond PixelNeRF and SparseFusion. For example, some subset of: 3DiM, Triplane Diffusion, Rodin, HoloDiffusion, NerfDiff, GeNVS. There is also the line of work considering distillation of pre-trained models, which is relevant because you use Stable Diffusion, such as:  NeuralLift-360 (Xu et al. 2022), NerDI (Deng et al. 2022), RealFusion (Melas-Kyriazi et al. 2023), Dreambooth-3D (Raj et al. 2023), Make-It-3D (Tang et al. 2023), Zero-1-to-3D (Liu et al. 2023).
* Some of these are mentioned in the introduction and include results on the same dataset as you (e.g. CO3D hydrants and ShapeNet Cars), but they are not included in the tables.
* To be clear, I would not expect the paper to compare to a large fraction of these methods, but I would expect the paper to compare to at least a few methods apart from PixelNeRF.

### Performance on ShapeNet Cars
* Along the lines of the comment above, the ShapeNet Cars table (Table 4 in the supplementary) shows relatively weak performance, but the table does not include many methods (including very old methods).  For example, {SRN, CodeNeRF, FE-NVS, VisionNeRF, 3DiM} all achieve better PSNR/LPIPS than the proposed model.

### Compute/Memory Utilization
* Most 3D-aware generative models tend to be compute-intensive (during training). It would be interesting to know how your model compares with others on these aspects, and how varying training time/data impacts results.

### [Minor] LaTeX Spacing
* It looks like some v-space tricks were applied (L181-182, L186-187, vertical padding on Table 2 and Table 3, all of Section 4.5). It’d be preferable to remove these.

**Questions:**

### Modeling the background
* You mention in Section 4.1 that you train on the hydrant category “incorporating the full background.” How exactly do you model the background? Did you consider any alternative approaches for this aspect of the problem?

**Limitations:**

The limitations are mentioned clearly at the end of the paper. This section suggests that they can be solved by “stronger geometry backbone and train[ing] it on larger datasets”. The question of whether or not this is all that is needed merits a longer discussion, but of course I understand that there are space limitations.

---

> ### Author Rebuttal · Authors · 2023-08-09
>
> Thank you for your time and effort sharing critical feedback regarding our work. We have addressed your points and questions about performance comparison and training costs below.
>
> > Comparison with other methods
>
> We agree with you and would like to open a discussion on concurrently proposed 3D novel view synthesis works, and respectively highlight that our method contributes different aspects compared to the majority of these.
>
> We would like to first clarify that our paper primarily contributes by 1) designing a framework to enable 2D frozen pre-trained diffusion model to perform novel-view synthesis without the need for time-consuming per-object test time distillation. 2) demonstrating that the utilization of a 2D prior in a pre-trained diffusion model can enhance the generalization capabilities of novel-view synthesis for unseen and open-set category objects.
>
> In order to substantiate our contributions, we would like to compare with concurrently proposed 3D novel view synthesis works, but considered three factors for baselines for a fair comparison:
>
> **1) the availability of open-source codebase** (but 3DiM and GeNVS do not provide, so we cannot compare with them fairly because we synthesize images with different resolutions),
>
> **2) methods that don't require laborious test-time optimization for each object** (a criterion that Nerdi, Dreambooth-3D, NeuralLift-360, RealFusion, and Make-It-3D do not fulfill),
>
> **3) image-conditional methods that can operate under sparse observations (1~3),**(Triplane Diffusion and HoloDiffusion are not designed for this).
>
> **4) Following your suggestion, we tried to compare our work with the concurrent Zero 1-to-3, however, we found that Zero 1-to-3 has strong data assumption:** it requires objects to be located at the origin and the placements of the cameras to be pointing towards the origin as a result of its synthetic 3D training data from Objaverse. CO3D, on the other hand, is a dataset of real-life video captures of objects with noisy camera trajectories where each frame does not ideally point at the center of the object. And thus, converting the camera view parameterization for compatibility with Zero 1-to-3 is not straightforward.
>
> Given that we've benchmarked our approach against the recently published SparseFusion (CVPR 2023) and GBT (arxiv 2023), which provides open-sourced codes, we believe our contributions have been suitably assessed in our setting. In this regard, our contributions remain unaffected regardless of comparisons with these baselines for other settings, and we would like to incorporate the thorough discussions with them into our revisions.
>
>
>
>
>
>
>
>
>
>
>
>
>
>
> > Performance on ShapeNet Cars
>
> This is a good point. This is mainly due to the following reasons. 1) PSNR results is low because of the sharpness of the results from diffusion model. In our response  PDF, we further add the results generated from our geometry model, we can see the PSNR score  improves by 10%. 2)  In our setting, we perform novel view synthesis on images with 512 resolution which is more challenging than the baselines you mentioned (they focus on  128 resolution). 3)  As described above, we focus on novel view synthesis without test time per-object optimization, a more challenging but practical setting for real-world deployment while others do not.
>
>
> > Compute/memory utilization
> Most 3D-aware generative models tend to be compute-intensive (during training). It would be interesting to know how your model compares with others on these aspects, and how varying training time/data impacts results.
>
> Our primary motivation is to enable a frozen 2D diffusion model to conduct novel view synthesis without the need for test-time optimization, so we only train the geometry module and the spatial guidance model within our framework, eliminating the need to fine-tune the entire diffusion model. This approach significantly improves our training efficiency. To provide a specific comparison, our framework requires only 3 days of training on eight A100-40GB GPUs, whereas the concurrent work, GeNVS, demands 7 days.
>
> Further, we provide the Figure about the relationship between training time and results in the PDF of response. The Figure clearly demonstrate xxx.
>
> > Background Modeling
>
> You mention in Section 4.1 that you train on the hydrant category “incorporating the full background.” How exactly do you model the background? Did you consider any alternative approaches for this aspect of the problem?
> Sorry for the confusing description. We do not design specific algorithms to model the background, and simply discard the foreground object mask during preprocessing to preserve the entirety of the scene including the background. Afterwards, the image goes through the same pipeline as the object-only scenario. The results demonstrate that the frozen pre-trained model can synthesize better novel view quality both object and scene level novel images when compared with models trained from scratch.
> In our paper, we focus on enabling the frozen 2D diffusion model to synthesize novel view without per-object test time optimization, and show its generalization ability on unseen category samples. In the future, we would like to add more conditions, e.g. segmentation map, for synthesizing more complex scene-level novel view synthesis.
>
>
> >  v-space tricks:
>
> Thank you for mentioning. We will remove them in the final version following your suggestions.

---

> > ### Author Response · Authors · 2023-08-14
> > **Sorry for the minor errors**
> >
> > We apologize for uploading an incorrect version of the response that contained one incomplete response. Here's the corrected sentence:
> >
> > >Compute/memory utilization Most 3D-aware generative models tend to be compute-intensive (during training). It would be interesting to know how your model compares with others on these aspects, and how varying training time/data impacts results.
> >
> > Our primary motivation is to enable a frozen 2D diffusion model to conduct novel view synthesis without the need for test-time optimization, **so we only train the geometry module and the spatial guidance model within our framework, eliminating the need to fine-tune the entire diffusion model.** This approach significantly improves our training efficiency. **To provide a specific comparison, our framework requires only 3 days of training on 8 A100-40GB GPUs, whereas the concurrent work, GeNVS, demands 7 days.**
> >
> > Further, we provide the Figure about the relationship between training time and results in the PDF of response. This figure clearly shows that as training steps increases, the performance of our method steadily improves.
> >
> >
> >
> > In summary, we thank the time and efforts the reviewers dedicated to our work, and we would like to revise paper following your suggestions. If you have further concerns, please feel free to respond us and we would like to discuss with you.

---

> > > ### Author Response · Authors · 2023-08-21
> > > **Additional experimental results**
> > >
> > > Thanks for your time and effort sharing feedback regarding our work.  Following your suggestions,  we perform additional comparisons with concurrent works:
> > >
> > > > Comparisons with GeNVS and 3DiM:
> > >
> > >
> > > To ensure a fair comparison with concurrent work GeNVS and 3DiM, which train diffusion models at a resolution of 128x128, we adapted our method to match this resolution for the intermediate feature map produced by our geometry module (original is 32 x 32 ). Nonetheless, our final image synthesis is at a higher resolution of 512x512. In contrast, both GeNVS and 3DiM only generate images at the 128x128 resolution. The detailed results are provided below:
> > >
> > >
> > > |                        | PSNR  | SSIM | LPIPS |
> > > |------------------------|-------|------|-------|
> > > | 3DiM                   | 21.01 | 0.57 | —     |
> > > | GeNVS (autoregressive) | 20.6  | 0.89 | 0.12  |
> > > | Ours                   | **21.31** | **0.89** | **0.12**  |
> > >
> > > ShapeNet Cars
> > >
> > >
> > >
> > > |       | PSNR  | SSIM | LPIPS |
> > > |-------|-------|------|-------|
> > > | GeNVS | 15.48 | 0.27 | 0.37  |
> > > | Ours  | **16.42** | **0.33** | 0.46  |
> > > CO3D Hydrant Category (object + background)
> > >
> > >
> > > From the results, it's evident that our method, **despite synthesizing images at a much higher resolution compared to the baselines (512 vs 128), still manages to outperform in certain metrics.** This is particularly evident in metrics such as PSNR and SSIM for hydrant scenes, as well as in the PSNR, SSIM, and LPIPS scores overall. Another significant advantage of our framework is its efficiency. It only necessitates 3 days of training on eight A100-40GB GPUs, while the competing method, GeNVS, takes twice as long, requiring a full 7 days.
> > >
> > > > Can you elaborate more on why converting camera poses is not straightforward? Couldn't you just use the relative poses provided by CO3D?
> > >
> > > The reasons why evaluating Zero 1-2-3 on CO3D dataset is not straightforward are listed as follows:
> > >
> > > 1. The location of a camera in Zero-1-2-3 is uniquely defined in a spherical coordinate system, which holds only under the assumption that the camera is always pointed at the center. Consequently, Zero 1-to-3 only parametrizes the relative camera pose by concatenating the change in polar angle, azimuth angle, and the radius(distance from the center) with respect to the given input view. However, cameras in CO3D are often pointing at different centers making the accurate calculation of relative polar and azimuth angles and radius with respect to a common center challenging.
> > >
> > > 2. All training assets in Zero 1-to-3 were normalized to fit within a unit cube with the camera distances from the center uniformly sampled in the interval [1.5, 2.2]. However, this is not strictly followed by CO3D cameras even after recentering and rescaling the scene.
> > >
> > > For better visualization of the difference in distribution of camera poses across the CO3D and Objaverse (as used by Zero 1-to-3) datasets, we have uploaded plots of the camera trajectories for several samples onto our project webpage.
> > >
> > > Regardless, to the best of our efforts, we tried to compute the relative CO3D poses that best estimate the parametrization of Zero 1-to-3 through recentering and rescaling of the poses and carried out novel view synthesis. **We have updated the project webpage with the results. It can be seen that Zero 1-to-3 struggles to preserve identity and consistency without test-time distillation, often deforming the input object.**
> > >
> > > In summary, we sincerely appreciate your suggestions for our paper, as they can significantly improve the quality of our work, and **we already perform additional experiments to compare with concurrent works in a more fair setting following your suggestions, and the results clearly demonstrate our superior performance over concurrent works even if we synthesise images with higher resolutions.
> > > We hope our responses sufficiently address your concerns regarding the results comparisons. And thus  we respectively hope you can reconsider your final decision. If you have further concerns, please feel free to respond to us and we would be happy to discuss with you.**

---

> > > ### Comment · Reviewer_fx5m · 2023-08-22
> > > **Thank you for your response**
> > >
> > > Thank you for your comprehensive response. I appreciate the effort that you put into your rebuttal, including your responses to the other reviewers. The additional experiments were helpful, especially the comparison to GeNVS. I will raise my rating from 5 to 6 and recommend acceptance.

---

### Author Rebuttal · Authors · 2023-08-10

Thanks for the time and effort sharing critical feedback of every reviewer regarding our work. o address the questions and points raised by the reviewers, we have provided additional experimental results in our response PDF.

1) In the hydrant-scene dataset, we've plotted a figure (Figure 1 in the response PDF) that demonstrates the consistent improvement in our method's performance over training time.

2) Our ablation study on the geometry module shows how our design enhances the overall performance. Notably, even in the absence of the new geometry model design, our framework outperforms the baselines.

3) We also provde addition qualitative results in our response PDF, which include both input images and images synthesized by our method.

4) **For more samples, please refer to the website linked in the abstract and the main paper.**

    a) Comparisons of synthesized 360-degree videos. **The results clearly show that our method can synthesise more consistent samples than baselines without test-time distillation.**

    b) Additional object-centric scene level 360-degree videos.

    c) Results with test-time distillation, highlighting improved consistency.

    d) Hydrant Scene Novel View Synthesis results with 5 context views.

In summary, we believe our work can bring different contributions from previous works for the community. Our work is centered on

1) how to utilize 2D prior from a frozen image diffusion model without test-time distillation.
2) how 2D prior from a **frozen** image diffusion can significantly bolster the generalization and efficacy of novel view synthesis.

Our results further unequivocally showcase the advantages of utilizing the 2D prior over the baseline methods. **Specifically, we achieve approximately a 50% improvement in FID and a 20% enhancement in LPIPS compared to open-sourced SparseFusion (CVPR 2023).** We believe our approach to utilizing the 2D prior for novel view synthesis can inspire more subsequent works (especially for open-set novel view synthesis without test time per-object distillation).

**We also provide specific responses to address concerns from each reviewer. We believe our detailed response addresses concerns about our method's performance and novelty. We kindly hope that you can consider our response in your final decision. If there are additional concerns or questions, we are happy to further discussion and would appreciate your feedback.**

Warm regards,

The Authors of Submission 12217

---

> ### Author Response · Authors · 2023-08-14
> **Sorry for the unnecessary grammatical errors**
>
> We apologize for uploading an incorrect version of the general response that contained grammatical errors. However, this does not affect the overall meaning. We sincerely hope for your understanding on this matter. We deeply regret any confusion it may have caused. If you have any further concerns or questions, we are always open for discussion and would truly appreciate your feedback.
>
> Best Regards,
>
> The Authors of Submission 12217

---

### Decision · Program_Chairs · 2023-09-21

**Decision:**

Accept (poster)

**Comment:**

The paper proposed a method for novel view synthesis from sparse views with the key idea being introducing a spatial guidance model (which is 3D consistent) for a frozen 2D diffusion model, such that the 2D diffusion model can generate novel view images (with a certain amount of 3D consistency) The results demonstrated a clear improvement over prior work (sparseFusion) both quantitatively and qualitatively.  The rebuttal successfully addressed the concerns of all the reviewers, and the key idea is a great contribution to the 3D community. Thus, I follow the consensus and recommend accepting this paper.

As mentioned by the reviewers and agreed by the authors, please tune down the claim on 3D consistency (in the introduction, Table 1 and corresponding experimental results), as the method does have the limitation of not guaranteeing 3D consistency.  Please also add the promised discussions and comparisons with other related works, as well as the clarification on the background modelling.